# Active particles driven by competing spatially dependent self-propulsion and external force

Lorenzo Caprini[1*], Umberto Marini Bettolo Marconi[2,3],
René Wittmann[1] and Hartmut Löwen[1]

**1** Heinrich-Heine-Universität Düsseldorf, Institut für Theoretische Physik II - Soft Matter,
D-40225 Düsseldorf, Germany
**2** Scuola di Scienze e Tecnologie, Università di Camerino,
Via Madonna delle Carceri, I-62032, Camerino, Italy
**3** INFN-Sezione di Perugia, I-06123, Perugia, Italy

★ lorenzo.caprini@hhu.de, lorenzo.caprini@gssi.it

## Abstract

We investigate how the competing presence of a nonuniform motility landscape and an external confining field affects the properties of active particles. We employ the active Ornstein-Uhlenbeck particle (AOUP) model with a periodic swim-velocity profile to derive analytical approximations for the steady-state probability distribution of position and velocity, encompassing both the Unified Colored Noise Approximation and the theory of potential-free active particles with spatially dependent swim velocity recently developed. We test the theory by confining an active particle in a harmonic trap, which gives rise to interesting properties, such as a transition from a unimodal to a bimodal (and, eventually multimodal) spatial density, induced by decreasing the spatial period of the self propulsion. Correspondingly, the velocity distribution shows pronounced deviations from the Gaussian shape, even displaying a bimodal profile in the high-motility regions. We thus show that the interplay of two relatively simple physical fields can be employed to generate complex emerging behavior.



# 1  Introduction

The control of active matter [1–4] is an important issue for technological, biological and medical applications and has recently stimulated many experimental and theoretical works. It is also very important in the future perspective of self-assembling and nano-fabricating active materials. The diffusivity of active particles is much higher than the one of their passive counterparts. Indeed, the former may be caused by high motility induced by either an internal "motor" (metabolic processes, chemical reactions, etc.) or a directed external driving force acting on each particle, while the latter is simply due to random collisions with the particles of the thermal bath. This property offers intriguing perspectives since it is possible to achieve navigation control of active particles [5,6], for instance when driving their trajectories by some feedback mechanism [7,8].

In the case of active colloids, such as Janus particles activated by external stimuli, the motility can be tuned by modulating the intensity of light [9–14]. This property has been employed to trap them [15,16] and to obtain polarization patterns induced by motility gradients [17,18]. Experimentally, the existence of an approximately linear relation between light intensity and swim velocity [19] allows to tune the motility and design spatial patterns with specific characteristics. Recent applications range from micro-motors [20,21] and rectification devices [22,23] to motility-ratchets [24]. Two experimental groups [25–27], have devised an intriguing technique to control the swimming speed of bacteria by using patterned light fields to enhance/reduce locally their motility by increasing/decreasing the light intensity. This leads to a consequent accumulation/depletion of particles in some regions, so that this procedure can be used to draw two dimensional images with the bacteria [25].

The fundamental physical concept behind experiments on light-controlled bacteria has been investigated many years ago in a theoretical context for noninteracting random walkers by Schnitzer [28] and later been extended to the interacting Run-and-Tumble model by Cates and Tailleur [29]: the lower the speed of active particles, the higher their local density. This theoretical result has been tested and confirmed in many numerical works and the existence of such a relation between particle velocity and density is now considered one of the most distinguishing features of active matter. A subsequent theoretical modeling of these effects has been proposed in Refs. [19,30–33] by generalizing the active Brownian particle (ABP)

model to include a spatially dependent swim velocity. This additional ingredient accounts for the well-known quorum sensing [34–36], chemotaxis and pseudochemotaxis [37–40] and correctly predicts a scaling of the density profile of individual particles with the inverse of the swim velocity. Including particle interactions in the ABP model may lead to the spontaneous formation of a membrane in two-step motility profiles [41] or cluster formation in regions with small activity [42]. Moreover, a temporal dependence in the activity landscape [43–48] may, in some cases, produce directed motion opposite to the propagation of the density wave [23, 49].

The ABP model has been widely employed to obtain theoretical predictions [50–52] and still represents one of the more spread active matter models for its versatility and broad applicability [53–58]. Nevertheless, the more recent active Ornstein-Uhlenbeck particle (AOUP) model [59–65], to be regarded as a "sister/brother" [66] of the ABP model, is generally easier to handle and often conveniently used to achieve further theoretical progress. For an ABP, the modulus of the active force is fixed and its orientation diffuses, while for an AOUP each component of the propulsion force evolves independently according to an Ornstein-Uhlenbeck process. Therefore, AOUPs can be used as an alternative to ABPs with simplified dynamics [67–70], to describe the behavior of a colloidal particle in an active bath [71, 72]. Moreover, a convenient mapping between the parameters of the two models can be performed on the level of the autocorrelation function of the self-propulsion velocity[1] [73], such that their predictions agree fairly well for small and intermediate persistence time of the active motion [66]. The present authors recently modified the AOUP model to account for a spatially dependent swim velocity in Ref. [74] and obtained exact results for both the density profile and velocity distribution of a potential-free particle.

Analytical results for active particles in competing external potential and motility fields are sparse. Therefore, in this work, we extend the theoretical treatment of free AOUPs with spatially dependent self-propulsion from Ref. [74] by including the presence of an external force field, revealing more interesting properties than those obtained for either a constant swim velocity or in the absence of external forces. For example, an AOUP with constant swim velocity displays a Gaussian density in a harmonic trap, while the competition between external forces and motility patterns gives rise to a density profile characterized by multiple peaks. Moreover, in the latter case, the velocity distribution at fixed position displays strong non-Gaussian effects and even a transition from a unimodal to a bimodal shape, which does not occur in the former case.

The paper is structured as follows: In Sec. 2, we present the model to describe an active particle in a spatially dependent swim-velocity landscape and subject to an external potential, while, in Sec. 3, we develop our theoretical approach to describe the steady-state properties of such a system. The theory is numerically tested in Sec. 4 in the case of a harmonic potential and sinusoidal swim-velocity profile. Finally, conclusions and discussions are reported in Sec. 5. The appendices contain derivations and information supporting the theoretical treatment.

---

[1]The common mapping between ABPs and AOUPs in $d > 1$ spatial dimensions relates the persistence time $\tau = D_r^{-1}/(d-1)$ and the active diffusion coefficient $D_a = v_0^2 \tau/d$ of the AOUP model to the rotational diffusivity $D_r$ and self-propulsion-velocity scale $v_0$ of the ABP model [73]. To ease the notation for the predictions of our generalized AOUP model, we follow the convention of Ref. [74] and do not imply this mapping, simply setting $d = 1$, which gives the same physics. If one wishes to make explicit contact to the ABP model for $d > 1$, the last term in Eq. (2b) should be replaced by $\sqrt{2\tau/d}\,\boldsymbol{\chi}$, such that the variance of the Ornstein-Uhlenbeck process at equal time becomes unity. Hence, the formulas subsequently derived for arbitrary spatial dimension $d$ should be interpreted by rescaling $u(\mathbf{x}) \to u(\mathbf{x})/\sqrt{d}$.

## 2 Model

### 2.1 Active particles with spatially dependent swim velocity

The position, $\mathbf{x}$, of an active particle evolves according to overdamped dynamics supplemented by a stochastic equation for the active driving, the so-called self-propulsion (or active) force, $\mathbf{f}_a$. Such a force term is responsible for the persistence of the trajectory and its physical origin depends on the system under consideration: flagella for bacteria and chemical reactions for Janus particles, to mention just two examples. The active force $\mathbf{f}_a$ can be written in the following form [66]:

$$\mathbf{f}_a = \gamma v_0 \boldsymbol{\eta}, \tag{1}$$

where $\boldsymbol{\eta}$ is a stochastic process of unit variance, $\gamma$ is the friction coefficient and $v_0$ is the constant swim velocity induced by the active force. To describe an active particle with a spatially dependent swim velocity, we employ the transformation $v_0 \rightarrow u(\mathbf{x}, t)$ in Eq. (1), which introduces a dependence on both position and time. The shape of $u(\mathbf{x}, t)$ must satisfy some properties related to physical arguments:

i) positivity: $u(\mathbf{x}, t) \geq 0$, for every $\mathbf{x}$ and $t$, since $u(\mathbf{x}, t)$ is the modulus of the velocity induced by the active force.

ii) boundedness: $u(\mathbf{x}, t)$ needs to be a bounded function of its arguments because the swim velocity cannot be infinite.

In what follows, we focus on the stochastic model introduced in Ref. [74], representing a generalization of the AOUP dynamics with $u(\mathbf{x}, t)$.

Assuming inertial effects to be negligible at the microscopic scale, typically realized at small Reynolds numbers, the overdamped dynamics of the active particle with spatially modulating swim velocity reads:

$$\gamma \dot{\mathbf{x}} = \mathbf{F} + \gamma \sqrt{2D_t} \boldsymbol{w} + \gamma u(\mathbf{x}, t) \boldsymbol{\eta}, \tag{2a}$$

$$\tau \dot{\boldsymbol{\eta}} = -\boldsymbol{\eta} + \sqrt{2\tau} \boldsymbol{\chi}, \tag{2b}$$

where $\boldsymbol{\chi}$ and $\boldsymbol{w}$ are $\delta$-correlated noises with zero average and unit variance and $\mathbf{F}$ is the force exerted on the particle, resulting from the gradient of a potential $U(\mathbf{x})$, i.e., $\mathbf{F}(\mathbf{x}) = -\nabla U(\mathbf{x})$. In this paper, we consider only a single particle, so that $U(\mathbf{x})$ is a one-body potential, but the description can be straightforwardly extended to the case of many interacting particles. The coefficient $D_t$ is the translational diffusion coefficient due to the solvent satisfying the Einstein's relation with $D_t = T_t/\gamma$ and the temperature, $T_t$, of the passive bath (for unit Boltzmann constant). The dynamics of $\boldsymbol{\eta}$ is characterized by the typical time, $\tau$, which represents the correlation time of the active force autocorrelation and is usually identified with the persistence time of the single-trajectory, i.e., the time that a potential-free active particle spends moving in the same direction with velocity $u(\mathbf{x}, t)$. In what follows, we neglect the contribution of the thermal bath by setting $D_t = 0$, which is well justified in most of the experimental active systems [1].

By writing the active force in Eq. (2) as $\mathbf{f}_a(\mathbf{x}, t) = \gamma u(\mathbf{x}, t) \boldsymbol{\eta}$ we have achieved two important goals. First, the spatial dependence of the self propulsion velocity can be conveniently accounted for through a multiplicative factor $u(\mathbf{x}, t)$, which does not affect the dynamics of the noise vector. Therefore, in the absence of external forces and time dependence, we analytically recover the law $\rho(\mathbf{x}) \sim 1/u(\mathbf{x})$ for the stationary density profile $\rho(\mathbf{x})$ [74]. Our model is, therefore, suitable to describe the class of experimental systems including engineered E. Coli bacteria [25, 27] or active colloids [19] for which this relation is observed. Note that the alternative spatially dependent AOUP dynamics proposed in Ref. [63] is characterized by equations

of motions that do not coincide with Eqs. (2) (as shown in the appendix of Ref. [74]). This alternative model thus leads to different predictions as it reproduces the relation $\rho(\mathbf{x}) \sim 1/u(\mathbf{x})$ only for slow spatial variation of $u(\mathbf{x})$ [63] but could be in principle suitable to describe another class of experiments. Second, the Ornstein-Uhlenbeck process in Eq. (2b) has unit equal-time variance (except for a dimensional factor which we ignore here for convenience, see footnote 1), such that the reduced stationary probability distribution of $\boldsymbol{\eta}$ does not depend on the time scale $\tau$. This means that $u(\mathbf{x})$ provides a unique velocity scale. Moreover, this unit-variance version of the AOUP model allows us to establish a direct link to the ABP model and an even larger family of models [66] for which Eq. (2a) has the same form. In other words, the relation $\rho(\mathbf{x}) \sim 1/u(\mathbf{x})$ in the force-free case can be consistently obtained for all dynamics of the self-propulsion vector $\boldsymbol{\eta}$. In turn, from Eq. (1) (or by taking $u(\mathbf{x}, t) = v_0$), the standard version of the AOUP model is recovered by absorbing the velocity scale $v_0$ into the active diffusion coefficient $D_{\mathrm{a}} = v_0^2 \tau$ (see also footnote 1 for a general discussion of the mapping between ABPs and the different versions of AOUPs in $d$ spatial dimensions). From this identification, we see that the condition, $D_{\mathrm{t}} \ll D_{\mathrm{a}}$, necessary to neglect the thermal noise requires $u(\mathbf{x}, t) > 0$, which is stronger than the one stated above.

## 2.2 Velocity description of an active Ornstein-Uhlenbeck particle (AOUP)

Our equation of motion (2a) of an AOUP with a spatially dependent swim velocity contains a multiplicative colored noise, which does not readily allow us to gain further analytic insight. As a first step to ease the theoretical treatment of our model, we switch to the auxiliary dynamics employed earlier in the potential-free case [74]. Instead of describing the system in terms of position $\mathbf{x}$ and self-propulsion velocity $u(\mathbf{x})\boldsymbol{\eta}$, we take advantage of the relation (holding for $D_{\mathrm{t}} = 0$)

$$\gamma \dot{\mathbf{x}} = \mathbf{F} + \gamma u(\mathbf{x}, t) \boldsymbol{\eta} \tag{3}$$

to perform the simple change of variables $(\mathbf{x}, \boldsymbol{\eta}) \rightarrow (\mathbf{x}, \dot{\mathbf{x}} = \mathbf{v})$. This trick allows us to directly study the position and the velocity of the active particle as for $u(\mathbf{x}, t) = v_0$. As in the potential-free case, to return to the original variables, we need to account for the space-dependent Jacobian matrix $\mathcal{J}$ of the transformation reported in Appendix A. The resulting Jacobian reads:

$$\det[\mathcal{J}] = u(\mathbf{x}, t), \tag{4}$$

where $\det[\cdot]$ represents the determinant of a matrix. Therefore, the probability distributions, $\tilde{p}(\mathbf{x}, \boldsymbol{\eta}, t)$ and $p(\mathbf{x}, \mathbf{v}, t)$, in the two coordinate frames satisfy the following relation:

$$\tilde{p}(\mathbf{x}, \boldsymbol{\eta}, t) = \det[\mathcal{J}] p(\mathbf{x}, \mathbf{v}, t). \tag{5}$$

Note that the condition $u(\mathbf{x}, t) > 0$ implies $\det[\mathcal{J}] > 0$ and, thus guarantees the possibility of performing the transformation. In what follows, we use these new variables to study a system subject to both a spatially dependent swim velocity, $u(\mathbf{x}, t)$, and an external potential $U(\mathbf{x})$. The generalization to include a thermal noise can be achieved by following Ref. [75].

To derive the dynamics in the variables $\mathbf{x}$ and $\mathbf{v}$, we adopt a simple strategy whose leading steps are reported in details in Appendix A. We perform the time-derivative of Eq. (3), substitute $\dot{\boldsymbol{\eta}}$ with Eq. (2b) and then replace $\boldsymbol{\eta}$ with $\mathbf{v}$ and $U(\mathbf{x})$, using again Eq. (3), and finally obtain an equivalent equation of motion for the velocity $\mathbf{v}$. The full result reads

$$\dot{\mathbf{x}} = \mathbf{v}, \tag{6a}$$

$$\gamma \tau \dot{\mathbf{v}} = -\gamma \mathbf{\Gamma}(\mathbf{x}) \cdot \mathbf{v} - \nabla U + \gamma u(\mathbf{x}, t) \sqrt{2\tau} \boldsymbol{\chi} \tag{6b}$$

$$+ \tau \frac{[\gamma \mathbf{v} + \nabla U]}{u(\mathbf{x}, t)} \left( \frac{\partial}{\partial t} + \mathbf{v} \cdot \nabla \right) u(\mathbf{x}, t).$$

In Eq. (6b), the first line is identical to the expression describing the constant case $u(\mathbf{x}, t) = v_0$: the dynamics of an overdamped active particle is mapped onto that of an underdamped passive particle with a spatially dependent friction matrix, $\gamma\mathbf{\Gamma}(\mathbf{x})$, which depends on the second derivatives of the potential and reads:

$$\mathbf{\Gamma}(\mathbf{x}) = \mathbf{I} + \frac{\tau}{\gamma}\nabla\nabla U(\mathbf{x}), \qquad (7)$$

where $\mathbf{I}$ is the identity matrix. Such a term increases or decreases the effective particle friction according to the value of the curvature of $U(\mathbf{x})$, which becomes more and more important as $\tau$ becomes large. In addition, as already found in the potential-free case, the noise amplitude contains a spatial and temporal dependence through the multiplicative factor $u(\mathbf{x}, t)$. The second line of Eq. (6b) contains the new terms, absent for $u(\mathbf{x}, t) = v_0$, accounting for both the time- and space-dependence of $u(\mathbf{x}, t)$.

For a further discussion of the new terms arising from a modulating swim-velocity profile, we restrict ourselves to the time-independent case, $u(\mathbf{x}, t) = u(\mathbf{x})$. Then, we identify two contributions to the total force. The first one, $\propto \mathbf{v}\mathbf{v}\cdot\nabla u$, is proportional to the square of the velocity and appears also in the absence of an external potential. Since it is even under time-reversal transformation, it cannot be interpreted as an effective Stokes force. The second force, $\propto (\nabla U)\mathbf{v}\cdot\nabla u$, couples the gradients of the potential and the swim velocity and gives rise to an extra space-dependent contribution to the effective friction. This allows us to absorb this term into a generalized effective friction matrix $\mathbf{\Lambda}(\mathbf{x})$, which reads:

$$\mathbf{\Lambda}(\mathbf{x}) = \mathbf{\Gamma}(\mathbf{x}) + \frac{\tau}{\gamma}\mathbf{F}(\mathbf{x})\frac{\nabla u(\mathbf{x})}{u(\mathbf{x})}, \qquad (8)$$

where $\mathbf{\Gamma}(\mathbf{x})$ is given by the expression for constant $u(\mathbf{x}) = v_0$ (see Eq. (7)). The new term in Eq. (8) linearly increases with increasing $\tau$ and provides a further spatial dependence to the friction matrix. Its sign is determined by $\nabla u(\mathbf{x})$ and $\mathbf{F}(\mathbf{x}) = -\nabla U(\mathbf{x})$, such that it can increase (positive sign) or decrease (negative sign) the effective friction. As a matter of fact, the spatial modulation of the swim velocity and the action of an effective potential are two distinct physical phenomena, which cannot be simply mapped onto each other. Indeed, $u(\mathbf{x})$ provides an additional contribution to the effective friction in the dynamics of $\mathbf{v}$ but does not give any contributions to the confining force, at variance with the potential $U(\mathbf{x})$ that affects both the force acting on the particle and the effective friction matrix[2] Moreover, the interplay between the gradients of both fields in the second term of Eq. (8) gives rise to nontrivial physical effects that will be investigated in the following.

---

[2]Here is an explicit example with a detailed discussion of the physical implications. To shed light on the essential physical difference between a confined particle with uniform swim velocity and a free particle subject to a swim-velocity profile, let us consider, as a basic example, an AOUP with constant swim velocity $u(\mathbf{x}) = v_0$ trapped in a harmonic potential, system (i), which can be solved exactly. The exact stationary density profile $\rho(\mathbf{x})$ of (i) has a Gaussian shape. In principle, this distribution can be realized also by a nontrivial swim-velocity profile in the absence of external forces, system (ii), upon choosing a modulation of the form $u(\mathbf{x}) \propto 1/\rho(\mathbf{x})$. However, the physics of (i) and (ii) are crucially distinct. In case (i), the particle is externally confined and can explore the region far from the minimum of the potential only because of fluctuations induced by the active force. In the case (ii), the particle is free and shows a diffusive behavior: the Gaussian density profile $\rho(\mathbf{x})$ is obtained since the particle spends more time in the central region where it moves slowly and because of the boundary conditions. More precisely, due to the absence of external forces (or other confining mechanisms), the swim velocity allows the particle to escape to infinity. This means that such an *effective* confinement can only formally be achieved through periodic boundary conditions: the particle moves slowly in the minimum of $u(\mathbf{x})$, escapes rightwards (or leftwards) with an increasing swim velocity and approaches again the slow region by coming back from the other side of the box. Dynamical observables like the mean-squared displacement are thus different in the two cases.

# 3 Theoretical predictions

## 3.1 Approximate stationary distributions

So far, all steps in Sec. 2.2 were exact and the drawn conclusions general. To make further theoretical progress, we continue to restrict ourselves to a static swim-velocity profile $u(\mathbf{x})$. At variance with the potential-free case, $U(\mathbf{x}) = 0$, the exact steady-state probability distribution of positions and velocities, $p(\mathbf{x}, \mathbf{v})$, is unknown and one needs to resort to approximations. To this end, we assume that all components of the probability current vanish, as in the case of a homogeneous swim velocity, $u(\mathbf{x}) = v_0$. As shown in Appendix B, this condition means that in the Fokker-Planck equation associated to Eq. (6) the effective drift and diffusive terms mutually balance. To derive a closed expression for the spatial density $\rho(\mathbf{x})$, we follow in Appendix B the idea of Hänggi and Jung behind the Unified Colored Noise Approximation (UCNA) [76–78]: having derived the auxiliary dynamics (6), where the colored noise $\boldsymbol{\eta}$ is replaced by a white noise $\boldsymbol{\chi}$, we formally identify a new variable $\dot{\mathbf{z}} := \dot{\mathbf{x}}/u(\mathbf{x})$ to eliminate the multiplicative nature of the noise and then neglect the generalized inertial term $\propto \ddot{\mathbf{z}}$ in Eq. (6b). This procedure yields an effective overdamped equation for the particle position $\mathbf{x}$ and finally, via the associated Smoluchowski equation for the time evolution of $\rho(\mathbf{x}, t)$, the stationary density distribution $\rho(\mathbf{x})$ for a system with space-dependent activity. Our theoretical method employs the vanishing-currents approximation and, as a consequence, allows us to derive an effective equilibrium theory whose validity will be investigated numerically. In the absence of thermal noise, the same (stationary) $\rho(\mathbf{x})$ can be obtained using the path-integral method proposed by Fox [79,80]. As already shown in the case of homogeneous swim velocity, both the UCNA and the Fox approach give rise to the exact distribution in the small-persistence regime, i.e., when the persistence time is smaller than the other typical times characterizing the dynamics, while they only capture the qualitative behavior of the system in the opposite regime (i.e., when the persistence time is comparable or larger than the other relevant time scales). In particular, in the present case, $\tau$ needs to be compared to both the relaxation time due to the potential and to the typical time induced by the spatial modulation of the swim-velocity profile (see Sec. 4.1 for an explicit discussion in the specific case of a harmonic oscillator).

Here, we report only the main results while the details of the derivation can be found in Appendix B. The whole stationary probability distribution reads:

$$p(\mathbf{x}, \mathbf{v}) \approx \rho(\mathbf{x}) \frac{\sqrt{\det[\mathbf{\Lambda}(\mathbf{x})]}}{\sqrt{2\pi} u(\mathbf{x})} \exp\left(-\frac{\mathbf{v} \cdot \mathbf{\Lambda}(\mathbf{x}) \cdot \mathbf{v}}{2 u^2(\mathbf{x})}\right). \tag{9}$$

We remark that the prefactor $\sqrt{\det[\mathbf{\Lambda}(\mathbf{x})]}/(\sqrt{2\pi} u(\mathbf{x}))$ is the explicit factor normalizing the conditional velocity distribution (i.e., at fixed position $\mathbf{x}$). The function $\rho(\mathbf{x})$ is approximated by

$$\rho(\mathbf{x}) \approx \frac{\mathcal{N}}{u(\mathbf{x})} \det[\mathbf{\Lambda}(\mathbf{x})] \exp\left(\frac{1}{\gamma\tau} \int^{\mathbf{x}} d\mathbf{y} \cdot \frac{\mathbf{\Lambda}(\mathbf{y}) \cdot \mathbf{F}(\mathbf{y})}{u^2(\mathbf{y})}\right), \tag{10}$$

with $\mathcal{N}$ being a normalization constant. Our expression for $\rho(\mathbf{x})$ coincides with the spatial density because it follows from integrating out the velocity in Eq. (9). The full distribution (9) displays a multivariate Gaussian profile in the velocity, whose covariance matrix accounts for the nontrivial coupling between velocity and position:

$$\langle \mathbf{v}\mathbf{v}(\mathbf{x}) \rangle = u^2(\mathbf{x}) \mathbf{\Lambda}^{-1}(\mathbf{x}). \tag{11}$$

The covariance $\langle \mathbf{v}\mathbf{v}(\mathbf{x}) \rangle$ is spatially modulated by $u(\mathbf{x})$, which also occurs in the potential-free case, so that, in the regions where the swim velocity is large, the particle moves faster.

Moreover, the external potential not only affects the velocity covariance through $\mathbf{\Gamma}(\mathbf{x})$, as in the case $u(\mathbf{x}) = v_0$ (see for instance Refs. [81,82]), but contains an additional spatial dependence through the coupling to the velocity gradient in the second term of Eq. (8).

We remark that a necessary condition to obtain predictions (9) and (10) (and, consequently, (11)) is that the matrix $\mathbf{\Lambda}$ is positive definite, so that its inverse exists. This is the main limitation of our theoretical approach, which is always suitable to describe the system in the small-persistence regime (when $\tau$ is small compared to the other relevant time scales), but can break apart in the large-persistence regime where the position-dependent part of the matrix $\mathbf{\Lambda}(\mathbf{x})$ becomes dominant (with respect to $\mathbf{I}$). As a consequence, our approach is supposed to work (at least qualitatively) in any spatial dimension and for every value of $\tau$ if (i) $U(\mathbf{x})$ is a convex function, (ii) $U(\mathbf{x})$ depends on a single Cartesian coordinate or has a positive slope in a radial geometry and (iii) the gradients of $U(\mathbf{x})$ and $u(\mathbf{x})$ enclose a sufficiently large angle $\alpha \geq \pi/2$. The requirements (i) and (ii) correspond to a positive definiteness of $\mathbf{\Gamma}(\mathbf{x})$, given in Eq. (7), and are thus already necessary for $u(\mathbf{x}) = v_0$ [73], while (iii) arises in addition from the second term in Eq. (8). If the two functions $U(\mathbf{x})$ and $u(\mathbf{x})$ violate either of the three conditions (i-iii), our predictions cannot qualitatively reproduce the behavior of a system for large enough $\tau$.

Since the distribution $\rho(\mathbf{x})$ from Eq. (10) can be interpreted as the effective density distribution of the system, the particle behaves as if it was subject to an effective potential, $\mathcal{V}(\mathbf{x}) := -\tau \gamma v_0^2 \ln(\rho(\mathbf{x}))$, which explicitly reads:

$$\mathcal{V}(\mathbf{x}) = -v_0^2 \int^{\mathbf{x}} d\mathbf{y} \cdot \frac{\mathbf{\Lambda}(\mathbf{y}) \cdot \mathbf{F}(\mathbf{y})}{u^2(\mathbf{y})} - \tau \gamma v_0^2 \ln\left( v_0 \frac{\det[\Lambda(\mathbf{x})]}{u(\mathbf{x})} \right), \tag{12}$$

up to a constant. This expression contains two terms, i) the spatial integral of the external force modulated by the inverse of the covariance matrix of the velocity distribution, cf. Eq. (11), and ii) the logarithm containing both the velocity modulation $u(\mathbf{x})$ and the determinant of the position-dependent matrix $\mathbf{\Lambda}^{-1}(\mathbf{x})$. For a constant swim velocity, $u(\mathbf{x}) = v_0$, we can perform the integral explicitly and the effective potential $\mathcal{V}(\mathbf{x})$ reduces to the known closed form found within the standard UCNA or Fox approach [73] since we neglect translational Brownian noise. Note that the spatial dependence of the swim velocity gives rise to an additional potential term with respect to the case $u(\mathbf{x}) = v_0$ contained in the expression for $\mathbf{\Lambda}(\mathbf{x})$. At equilibrium, when $u(\mathbf{x}) = v_0$ and $\tau \to 0$, the density reduces to the well known Maxwell-Boltzmann profile, since $\mathbf{\Lambda}(\mathbf{x})$ becomes unity.

Finally, we remark that there exists a formal mapping of a system subject to a spatially dependent swim velocity, represented by a multiplicative noise in Eq. (2a), to a system with an effective external potential and additive noise. This mapping is mediated by a change of variables, reported and discussed in appendix C for completeness. Such a trick has been proven particularly helpful to find the explicit solution in a one-dimensional Run&Tumble model without external potential [83]. In our case, the newly defined variable allows us to identify the proper acceleration term which should be neglected to obtain the final UCNA result. However, to describe physically meaningful coordinates, i.e., the position and velocity of an AOUP, it is imperative to account for the notion of a swim velocity field $u(\mathbf{x})$ in the stationary distributions of an AOUP stated above.

## 3.2 Multiscale method for the full-space distribution

To check the validity of our predictions, at least in the small-persistence regime, we resort to an exact perturbative approach in powers of the persistence time $\tau$. For simplicity, the technique is presented in the one-dimensional case because the generalization to higher dimensions is technically more involved and does not provide additional insight. In addition, as in experiments based on active colloids [19], we will consider a one-dimensional swim-velocity profile

$u(x)$ in the remainder of this work, justifying our particular attention to the one-dimensional case in the following presentation.

Our starting point is the following Fokker-Planck equation for the probability distribution $p(x, v, t)$:

$$\partial_t p = \frac{\Lambda(x)}{\tau}\frac{\partial}{\partial v}(vp) + \frac{u^2(x)}{\tau}\frac{\partial^2}{\partial v^2}p - \frac{F(x)}{\tau\gamma}\frac{\partial}{\partial v}p - v\frac{\partial}{\partial_x}p - \frac{1}{u(x)}\left(\frac{\partial}{\partial x}u(x)\right)\frac{\partial}{\partial v}\left(v^2 p\right), \quad (13)$$

associated to the dynamics (6) in one spatial dimension. Its solution is unknown for a general potential $U(x)$, even in the special case $u(x) = v_0$. Therefore, one needs to resort to approximation methods or perturbative strategies to obtain analytical insight. As shown in previous works [84,85], it is possible to obtain perturbatively both the full distribution $p(x, v, t)$ and the configurational Smoluchowski equation for the reduced space distribution $\rho(x, t)$ following the method developed by Titulaer in the seventies [86]: starting from the Fokker-Planck equation (13) the velocity degrees of freedom can be eliminated by using a multiple-time-scale technique. Physically speaking, the fast time scale of the system corresponds to the time interval necessary for the velocities of the particles to relax to the configurations consistent with the values imposed by the vanishing of the currents. The characteristic time of the slow time scale is much longer and corresponds to the time necessary for the positions of the particles to relax towards the stationary configuration.

In the present case, the perturbative parameter is the persistence time $\tau$. Since we are mainly interested in time-independent properties, we limit ourselves to compute the steady-state probability distribution by generalizing the results of Refs. [87,88] previously obtained for the case $u(x) = v_0$ (see also Ref. [89] for a more general expansion with an additional thermal noise). For space reasons, the details of the calculations are reported in Appendix D. Our main result is the following exact perturbative expansion of the distribution $p(x, v)$ in powers of the parameter $\tau$:[3]

$$
\begin{aligned}
p(x, v) = \rho_s(x)p_s(x, v)&\left\{1 + \frac{\tau}{\gamma}\left[\frac{1}{2}U''(x) - \frac{v^2}{2u^2(x)}U''(x) + \left(\frac{v^2}{2u(x)^2} - \frac{1}{2}\right)U'(x)\frac{u'(x)}{u(x)}\right]\right.\\
&\left. + \frac{\tau^2}{6\gamma}u(x)\left(\frac{v^3}{u^3(x)} - 3\frac{v}{u(x)}\right)\left[U'''(x) - \frac{\partial}{\partial x}\left(\frac{U'(x)}{u(x)}u'(x)\right)\right]\right\} + O(\tau^3),
\end{aligned}
\quad (14)
$$

where the normalized distribution $p_s(x, v)$ is given by

$$p_s(x, v) = \frac{\mathcal{N}}{\sqrt{2\pi}u(x)}\exp\left(-\frac{v^2}{2u^2(x)}\right), \quad (15)$$

and the function $\rho_s(x)$ reads

$$\rho_s(x) = \mathcal{N}\frac{\Lambda(x)}{u(x)}\exp\left(-\frac{1}{\gamma\tau}\int^x dy\, U'(y)\frac{\Lambda(y)}{u^2(y)}\right), \quad (16)$$

with the normalization factor $\mathcal{N}$ and the prime as a short notation for the spatial derivative. Already at order $\tau/\gamma$ our general result (14) for a nonuniform swim velocity contains an extra term proportional to $\partial_x u(x)$, compared to the expansion derived in Ref. [87,88] (see footnote 3), which is responsible for an additional coupling between position and velocity.

The product $\rho_s(x) \times p_s(x, v)$ in Eq. (14) plays the role of an effective equilibrium-like distribution, which is exact in the limit $\tau \to 0$. The required expression (15) for $p_s(x, v)$ is the

---

[3]Notice the slightly different role of the expansion parameter $\tau$ in Eq. (14) for our version of the AOUP model (2), in which we have eliminated the active diffusion coefficient $D_a \propto \tau$ in favor of an expression containing the explicit factor $\tau$.

exact solution of the potential-free active system with a spatially dependent swim velocity as derived in Ref. [74]: it is a Gaussian probability distribution for the particle velocity $v$ with an effective position-dependent kinetic temperature provided by $u^2(x)$. The spatial density $\rho_s(x)$ from Eq. (16) corresponds to the UCNA result (10) in one dimension. Our previous approximated expression (9) for $p(x, v)$ is consistent with the full result (14) at first order in the expansion parameter $\propto \tau$. The first deviation between the two formulas occurs at order $O(\tau^2)$, where the exact expression for $p(x, v)$ contains additional odd terms in $v$. The exact density profile $\rho(x) = \int \mathrm{d}v p(x, v) = \rho_s(x) + O(\tau^2)$ deviates from the UCNA result beyond linear orders in $\tau$. As a consequence, we expect that our UCNA approximation, $\rho_s(x)$, is exact in the small-persistence regime, while it could only reproduce the qualitative behavior of $\rho(x)$ in the large-persistence regime.

# 4 The harmonic oscillator

## 4.1 Swim-velocity profile and external potential in one dimension

In this section, we present and investigate the interplay between a spatially modulated swim velocity and an external confining potential in one spatial dimension. While, in Sec. 3.2, we have shown that our analytical predictions from Eqs. (9), (10) and (11) are exact in the small-persistence regime through analytical arguments, a numerical analysis is necessary to check our approximations in the large-persistence regime.

To fix the form of the profile $u(x)$ employed in our numerical study and theoretical treatment, we take inspiration from experimental works on active colloids [19] and consider a static periodic profile $u(x)$ varying along a single direction, namely the $x$ axis, so that:

$$u(x) = v_0 \left( 1 + \alpha \cos \left( 2\pi \frac{x}{S} \right) \right), \tag{17}$$

where $\alpha < 1$ and $v_0 > 0$ so that $u(x) > 0$ for every $x$. The parameter $\alpha$ determines the amplitude of the swim velocity oscillation while $S > 0$ sets its spatial period. As a consequence, the active particle is subject to the minimal swim velocity $v_0(1 - \alpha)$ and to the maximal one $v_0(1 + \alpha)$. This choice and the features of the AOUP allow us to consider directly a one-dimensional system, essentially focusing only on the $x$ component and neglecting the dynamics of the other spatial coordinates.

We remind that, in the potential-free case [74], the system admits two typical length scales, i.e., the persistence length $v_0\tau$ and the spatial period, $S$, of the swim-velocity profile (17). In other words, by rescaling the time by $\tau$ and the particle position by $v_0\tau$, the dynamics is controlled by the dimensionless parameter $v_0\tau/S$ and by the dimensionless parameter $\alpha$ quantifying the amplitude of the swim-velocity oscillation. The external force $F(x)$ then introduces at least one additional length-scale, $\ell$, which depends on the specific nature of $F$, and, thus, an additional dimensionless parameter, say $\ell/(v_0\tau)$, related to the external potential. The last dimensionless parameter controls the dynamics also in the case $u(x) = v_0$ [90]. Now, we can identify the small-persistence regime, where the self-propulsion velocity relaxes faster than the particle position, with the criterion $v_0\tau/S \ll 1$ and $\ell/(v_0\tau) \gg 1$. Under the former condition, we expect that the system behaves as its passive counterpart: if $\tau \ll S/v_0$ holds, the self-propulsion behaves as an effective white noise. In the opposite case, when $v_0\tau/S \gg 1$, the dynamics is strongly persistent and we expect intriguing nonequilibrium properties.

To proceed further with the numerical investigation, we consider a simple shape for the confinement, i.e., the harmonic potential

$$U(x) = \frac{k}{2} x^2, \tag{18}$$

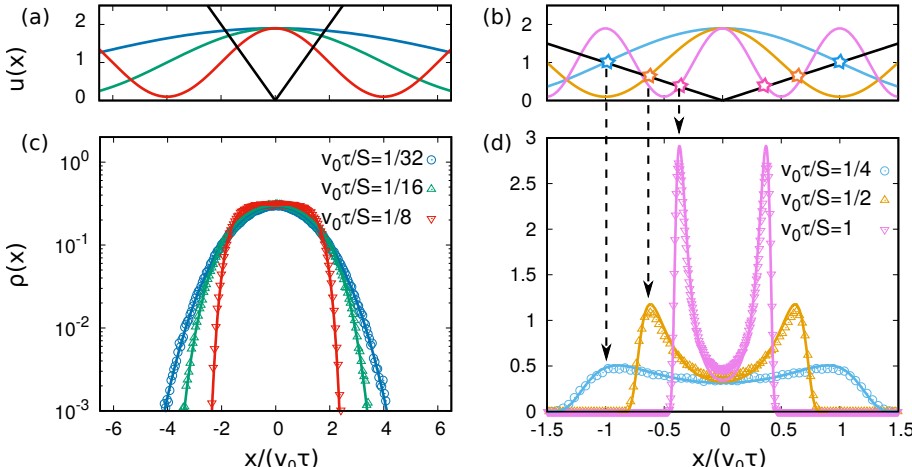

Figure 1: Density distributions. Panels (a) and (b): swim-velocity profile $u(x)$ for different values of $S$ (colored curves), compared with the modulus of the linear force profile, $|F(x)| = -k|x|/\gamma$ (black curve) as a reference. The colored stars are placed at the first cross point between $u(x)$ and $F(x)/\gamma$. Panels (c) and (d): spatial density profile, $\rho(x)$, for different values of $S$. Points are obtained by numerical simulations while solid lines by plotting the theoretical prediction (10) (that reduces to Eq. (16) in one dimension with $\Lambda(x)$ given by Eq. (19)). The integral occurring in Eq. (16) has been performed numerically. Panels (a), (c) and (b), (d) share the same legend. Simulations are realized in one spatial dimension with $\tau k/\gamma = 1$ and $\alpha = 0.9$.

where the constant $k$ determines the strength of the linear force. The dimensionless parameter associated with this external potential is thus $k\tau/\gamma$, i.e., $\ell = v_0\tau^2 k/\gamma$. By observing that the curvature of the potential is constant, the effective friction coefficient $\Lambda(x)$ from Eq. (8) becomes:

$$\Lambda(x) = \left(1 + \tau\frac{k}{\gamma}\right)\left(1 + \frac{\alpha x}{u(x)^2}\frac{2\pi}{S}\sin\left(2\pi\frac{x}{S}\right)\frac{\tau\frac{k}{\gamma}}{1 + \tau\frac{k}{\gamma}}\right). \tag{19}$$

As shown by Eq. (19), the two dimensionless parameters $\tau k/\gamma$ and $\alpha$ play a similar role. Indeed, they only determine the relative amplitude of the spatial modulation of $\Lambda(x)$. When either $\alpha$ or $\tau k/\gamma$ vanish, the effective friction becomes constant and the coupling between velocity and position disappears. Instead, when we approach both limits $\tau k/\gamma \to \infty$ and $\alpha \to 1$, the amplitude of the spatial oscillations becomes maximal. By varying the dimensionless parameter $v_0\tau/S$, on the other hand, one can explore the different properties of the system: when $v_0\tau/S$ grows, the spatial period of $u(x)$ decreases and the position-dependent term of $\Lambda(x)$ becomes less relevant. To study the resulting behavior of the system in detail, we keep fixed $\alpha = 0.9$ and $\tau k/\gamma = 1$ and we change only $v_0\tau/S$ to study the properties of the system.

## 4.2 Density distribution

Before considering the peculiar behavior of an AOUP with spatial-dependent swim velocity and confined in a harmonic trap (18), it is useful to remind the reader that, in the absence of motility landscape, the AOUP in a harmonic trap is described by a Gaussian density. The spatial density profile, $\rho(x)$, are shown in Figs. 1 and 2 for the spatial profile of $u(x)$ given by Eq. (17). The bottom panels show $\rho(x)$ for different values of the spatial period $S$ (through the dimensionless parameter $v_0\tau/S$) of the swim velocity $u(x)$, which we compare in the top panels to the modulus $|F(x)|/\gamma$ of the confining force $F(x) = -U'(x)$ as a reference.

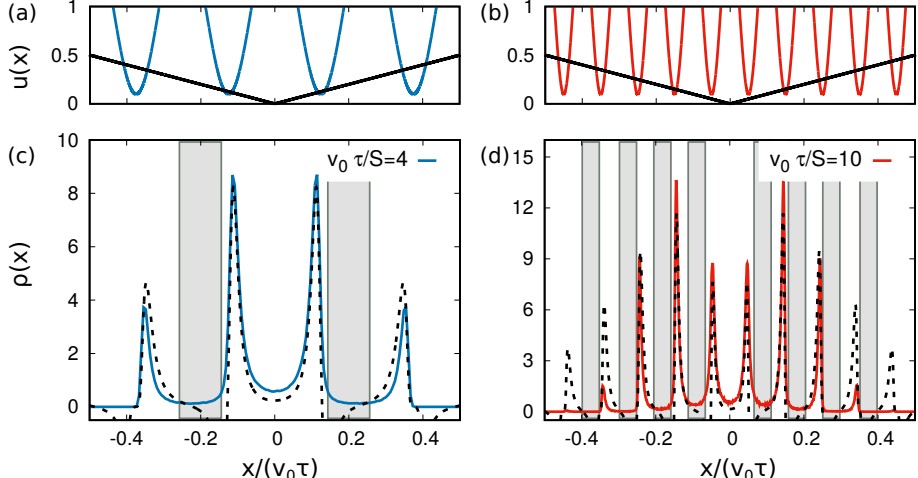

Figure 2: Density distributions. Panels (a) and (b): swim-velocity profile $u(x)$ for two different values of $S$ (colored curves), compared with the modulus of the linear force profile, $|F(x)| = -k|x|/\gamma$ (black curve) as a reference. Panels (c) and (d): spatial density profile, $\rho(x)$, for different values of $S$. Solid colored lines (blue in panel (c) and red in panel (d)) are obtained by numerical simulations while dashed black lines by plotting the theoretical prediction (10) (that reduces to Eq. (16) in one dimension with $\Lambda(x)$ given by Eq. (19)). The integral occurring in Eq. (16) has been performed numerically. Grey rectangles are drawn in the regions where Eq. (10) is not defined, say when $\Lambda(x) < 0$ (see the main text for more details). Panels (a), (c) and (b), (d) share the same legend. Simulations are realized in one spatial dimension with $\tau k/\gamma = 1$ and $\alpha = 0.9$.

In the small-persistence regime, $v_0\tau/S \ll 1$ (see panels (a) and (c) of Fig. 1), the unimodal density distribution is fairly described by expanding the UCNA solution (10) in powers of $x/S$, obtaining:

$$\rho(x) \sim \exp\left(-\frac{1+\tau\frac{k}{\gamma}}{(1+\alpha)^2}\frac{k}{v_0^2}\frac{x^2}{2}\right). \qquad (20)$$

In the expression (20), we have neglected the terms proportional to $x^2/S^2$, $x^4/S^2$ and all higher-order terms in power of $\sim 1/S$. Remarkably, even in this crude approximation, we see from the factor $(1+\alpha)^2$ that the oscillations of the swim velocity lead to a decrease of the second moment $\langle x^2 \rangle$ of $\rho(x)$ compared to the homogeneous case $u(x) = v_0$. This prediction is consistent with previous results obtained in the absence of an external potential, where the swim-velocity oscillations produce the decrease of the long-time diffusion coefficient [74] (see also Ref. [91]). In this regime, the spatial pattern $u(x)$ produces an effective potential with increasing stiffness for increasing spatial modulation. For higher $v_0\tau/S$, the distribution starts developing non-Gaussian tails, which are still well-described by including higher-order terms in the UCNA expansion (20).

When increasing $v_0\tau/S$ further (see panels (b) and (d) of Fig. 1), $\rho(x)$ becomes a bimodal distribution with two peaks symmetric to the origin, as in a system confined in a double-well potential. This effect is absent in the case $u(x) = v_0$ where the AOUP density distribution in a harmonic potential always has a Gaussian shape [60, 92–94]. For a position-dependent swim velocity, the comparison between the analytical result (10) (that reduces to Eq. (16) in one dimension) and the numerical simulations still reveals a good agreement: in particular, Eq. (10) is able to predict the observed bimodality of the distribution. To explain the occurrence of this

bimodality in the shape of $\rho(x)$, we can use an effective (but rather general) force-balance argument in Eq. (2a). This argument can be applied to the present intermediate-persistence regime, $v_0\tau/S \sim 1$ (or also for $v_0\tau/S \gg 1$ discussed later), where the self-propulsion vector $\eta$ in the active force can be considered to be roughly constant for typical times $t \lesssim \tau$. Since the variance of $\eta$ is unitary, the most likely value assumed by the self-propulsion velocity at point $x$ is simply $u(x)$ (in absolute value). For this reason, it is generally unlikely to find the particle in regions with $u(x) < |F(x)|/\gamma$, because there the particle's self propulsion is not sufficient to climb up the potential gradient. Moreover, in the spatial points where $u(x) > |F(x)|/\gamma$, the active particle does not get stuck on average because its high self-propulsion velocity allows for its directed motion until $u(x) = |F(x)|/\gamma$ is fulfilled. When this force balance occurs, the particle can explore further spatial regions only because of large (and rare) fluctuations of $\eta$. This reasoning is confirmed by inspecting Fig. 1 for different fixed values of $v_0\tau/S$. It is evident from the dashed arrows that the peaks of the distribution in Fig. 1(d) coincide with the intersection between the modulus $|F(x)|/\gamma$ of the external force (black curve) and $u(x)$ (colored curves) in Fig. 1(c).

Starting from the theoretical result (10), we can predict the critical value $S_c$ at which the distribution becomes bimodal, by simply requiring that $\mathrm{d}^2/\mathrm{d}x^2\rho(x) = 0$ (at $x = 0$), obtaining:

$$\frac{S_c^2}{v_0^2\tau^2} = (2\pi)^2\alpha(1+\alpha)\left[\frac{1+3\tau\frac{k}{\gamma}}{\left(1+\tau\frac{k}{\gamma}\right)^2}\right]\frac{\gamma}{k\tau}. \tag{21}$$

In general, we predict that the value of $S_c/(v_0\tau)$ increases with increasing $\alpha$ (recall that $0 < \alpha < 1$) and is a decreasing function of $\tau k/\gamma$. This is consistent with our physical intuition: larger oscillations (i.e., larger $\alpha$) facilitate the transition to a bimodal shape. Indeed, the larger $\alpha$, the smaller the minimal self-propulsion velocity, that hinders the particle's ability to come back to the origin. Instead, the increase of $\tau k/\gamma$ gives rise to the opposite behavior: the larger $\tau k/\gamma$, the steeper the effective confining trap. As a consequence, the active particle needs larger fluctuations of $\eta$ to reach spatial regions where $u(x)$ assumes low values which compete with the external force. Specifically, for the chosen parameters $\alpha = 0.9$ and $\tau k/\gamma = 1$, Eq. (21) predicts the onset of bimodality for $v_0\tau/S > 1/8$. From Fig. 1, we also observe that the increase of $v_0\tau/S$ beyond this threshold enhances the bimodality showing two symmetric peaks with increasing height but occurring at spatial positions which get closer.

In the large-persistence regime $v_0\tau/S \gg 1$ (see Fig. 2), we observe the emergence of many symmetric peaks in $\rho(x)$. Their positions are still determined by the balance between $u(x)$ and $|F(x)|/\gamma$, and, in this case, roughly coincide with the minima of $u(x)$ close to the origin (i.e., the minimum of $U(x)$). As shown in Fig. 2 (a), $u(x)$ first crosses $|F(x)|/\gamma$ almost in its first minima (at $x/v_0\tau \approx \pm0.15$) for $v_0\tau/S = 4$. This implies that small fluctuations of the self-propulsion velocity allows the particle to explore spatial regions which are even more distant from the potential minimum, so that it also accumulates at the second crossing point (at $x/v_0\tau \approx 0.4$). According to Fig. 2 (c), the height of these secondary peaks is smaller than that of the primary ones because the particle remains trapped at the first balance points for most of the time, while only on rare occasions its swim velocity is sufficient to further climb up the potential gradient. In Fig. 2 (d), for an even larger value of $v_0\tau/S = 10$, we observe that the height of the peaks near the origin is lower than that of the successive peaks. In this case, Fig. 2 (b) shows that the minima of $u(x)$ closest to the origin are still larger than $|F(x)|/\gamma$, so that (most of the time) the particle has a sufficiently large self-propulsion velocity to go further until entering the spatial region where the first intersection of $u(x)$ and $|F(x)|/\gamma$ occurs. We conclude that, even in the case of a harmonic potential, the oscillation of the swim velocity allows the AOUP to climb up the potential barrier and accumulate preferably in spatial regions (corresponding to minima of $u(x)$), which are further away from the origin.

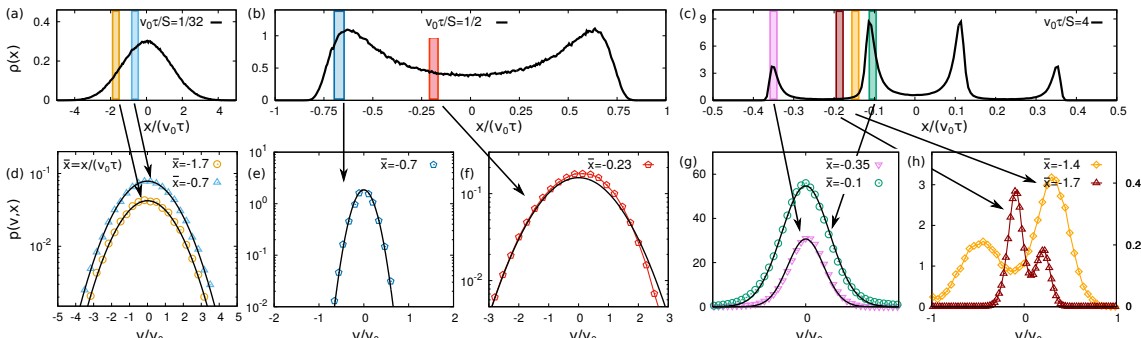

Figure 3: Velocity distributions. Panels (a), (b) and (c): simulated density distribution $\rho(x)$ for $S/(v_0\tau) = 32, 2, 1/4$, respectively, as a reference. Colored rectangles are drawn in correspondence of the spatial regions used to calculate the velocity distribution in the other panels. Panels (d), (e), (f), (g) and (h): velocity distribution $p(v,x)$ as a function of $v$ calculated at fixed positions $x = \bar{x}v_0\tau$ according to the legend. Panel (d) is calculated at $S = 32$, panels (e) and (f) at $S = 2$ and panels (g) and (h) at $S = 0.25$. Colored symbols and lines are obtained by numerical simulations and solid black lines show the theoretical prediction (9) if applicable (the theory yields a one-dimensional Gaussian and requires a positive definiteness of $\Lambda(x)$ given by Eq. (19)). Simulations are realized in one spatial dimension with $\tau k/\gamma = 1$ and $\alpha = 0.9$.

Finally, we note that in the large-persistence regime, the UCNA prediction (10) (or Eq. (16) in one-dimension) for the spatial distribution fails. This occurs because of the presence of spatial regions where the effective friction $\Lambda(x)$, given by Eq. (19), becomes negative (see the gray-shaded regions in Fig. 2(c) and (d)). This implies that also the corresponding approximation for $\rho(x)$ can assume negative values. This failure resembles the one of the UCNA (or the Fox approach) for the standard AOUP model with $u(x) = v_0$ confined in a nonconvex potential [73]. In that case, the strongly non-Gaussian nature of the system is at the basis of new intriguing phenomena, such as the occurring of effective negative mobility regions [95], the overcooling of the system [96] and the violation of the Kramers law for the escape properties [97, 98]. We expect that our model could display a similar phenomenology and that such problems can be treated by using similar theoretical techniques [95, 99–101]. However, we stress that the generalized UCNA still accurately predicts the positions of the main peaks in Fig. 2 (c), although in Fig. 2 (d) there emerge additional smaller peaks further away from the origin, which are absent in simulations. The appearance of those fake peaks is reminiscent of the overestimated wall accumulation predicted by UCNA for $u(x) = v_0$.

## 4.3 Velocity distribution

A potential-free AOUP in a motility landscape and an AOUP with constant swim velocity confined by harmonic traps are both described by Gaussian distributions of the velocity. To illustrate the new properties arising from the interplay between these two fields, we focus on the dependence of the full joint probability density $p(x,v)$ on the velocity, shown in Fig. 3 for some representative values of the particle's position $x$. Moreover, we choose three different values of $S/(v_0\tau)$ to explore the three distinct regimes observed in Sec 4.2. For each regime, we report once again the density distribution $\rho(x)$ in panels (a), (b) and (c), where colored bars mark the regions for which we calculate $p(x,v)$ as a function of $v$ in panels (d), (e), (f), (g), (h).

In the regime of small persistence, $(v_0\tau)/S \ll 1$, the shape of $p(v,x)$ is Gaussian, inde-

pendently of the position $x$ (Fig. 3 (d)). This result fully agrees with the prediction (9) (that simply reduces to a one-dimensional Gaussian) as revealed by the comparison between colored data points and black solid lines in Fig. 3 (d). As predicted by the position-dependent variance in Eq. (11), different positions $x$ come along with a change in the width of the velocity distribution.

In Fig. 3 (e) and (f), the regime of intermediate persistence, $v_0\tau/S \sim 1$, is investigated, which displays a bimodality in the density distribution. Here, we compare $p(x,v)$ calculated in the vicinity of a peak of $\rho(x)$ to the velocity profile near the local minimum of $\rho(x)$ (close to the origin). In the former case, the distribution $p(x,v)$ displays an almost Gaussian shape in agreement with Eq. (9), while in the latter case, it deviates from the theoretical prediction due to its non-Gaussian nature. In particular, the shape of $p(x,v)$ becomes asymmetric in $v$ and develops non-Gaussian tails. While the prediction (9) cannot account for the non-Gaussianity induced by the interplay of confinement and spatially modulating swim velocity, we remark that its quality near the regions where the particle preferably accumulates is still very good. This conclusion resembles the one obtained in Ref. [90], where an AOUP (with $u(x) = v_0$) has been studied in a single-well anharmonic confinement.

Finally, the large-persistence regime, $v_0\tau/S \gg 1$, where the density distribution has multiple peaks also gives rise to a rich phenomenology of the stationary velocity profile, as shown in Fig. 3 (g) and (h). In the spatial regions for which $\Lambda(x) > 0$, i.e., where the particles accumulate, the velocity distribution $p(x,v)$ (at fixed $x$) is again well described by the Gaussian distribution with position-dependent variance given by Eq. (9) (see Fig. 3 (g)), as in the case $v_0\tau/S \lesssim 1$. Instead, in the spatial regions where $\Lambda(x) < 0$, i.e., between the primary and the secondary peaks (see also Fig. 2 (c)), the distribution displays a non-Gaussian shape (see Fig. 3 (h)). Compared to the case $v_0\tau/S \sim 1$, the non-Gaussianity is much more evident due to the occurrence of a bimodal behavior in the velocity distribution. In more detail, upon shifting the coordinate $x$ in the first argument of $p(x,v)$ closer to the origin (Fig. 3 (g) and (h)), we observe that, starting from a nearly Gaussian shape centered at $v = 0$ (pink curve), the main peak moves toward $v < 0$ and a second small peak starts growing for $v > 0$ (brown curve). Shifting again $x$, the second peak becomes dominant (yellow curve) and moves closer toward $v = 0$ until the distribution is again described by a Gaussian (green curve). . This phenomenology resembles the one observed in the case of an AOUP with $u(x) = v_0$ in a double-well potential [95]. Also in the latter case, the velocity distribution at fixed position exhibits bimodality in the spatial regions where the effective friction coefficient $\Lambda(x) \simeq \Gamma(x)$ becomes negative, although this effect is then induced by the negative curvature of the potential. Intuitively, particles that are stuck in an accumulation region placed far from the potential minima (where $u(x)\eta$ balances the confining force) could move back towards the center or other minima when their active force varies because of the noise fluctuations. For example, when $\eta$ changes sign (or $|\eta|$ decreases), the particle comes back leftward or rightward (depending on the sign of $\eta$) moving with a large velocity induced by the deterministic force (that is particularly large far from the potential minimum) until to reach a new accumulation region. This simple argument provides an additional intuitive explanation for the bimodality of the velocity distribution.

## 4.4 Spatial profile of the kinetic temperature

To emphasize the dynamical effects due to the spatial modulation of the swim velocity, we focus on the profile of the kinetic temperature defined as the variance of the particle velocity, $\langle v^2(x)\rangle$. We show $\langle v^2(x)\rangle$ as a function of $x$ in Fig. 4 for values of $v_0\tau/S$ spanning all regimes from small (panel (c)) to intermediate and large persistence (panel (d)), see also panels (a) and (b) for a direct comparison with the corresponding density profiles.

For small values of $v_0\tau/S \ll 1$, the spatial profile of the variance, $\langle v^2(x)\rangle$, is rather flat and attains its maximum value at $x = 0$, i.e., the position of the potential minimum (see Fig. 4 (c)).

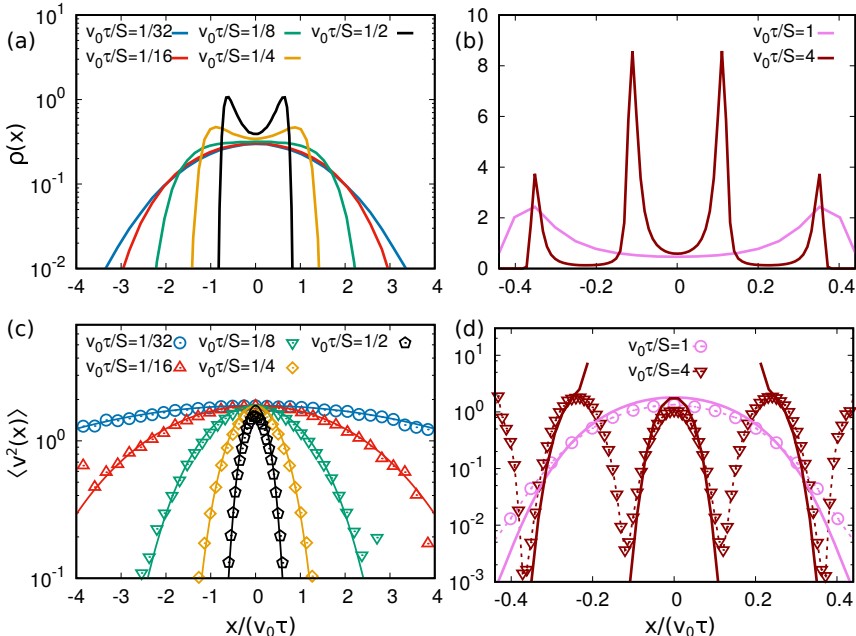

Figure 4: Spatial profiles of the kinetic temperature. Panels (a) and (b): simulated density profile $\rho(x)$ for different values of the dimensionless parameter $v_0\tau/S$ as a reference. Panels (c) and (d): kinetic temperature $\langle v^2(x)\rangle$ with the same color legend. In particular, panels (a) and (c) show the small-persistence regime, $v_0\tau/S \leq 1$, while panels (b) and (d) display the intermediate-persistence and large-persistence regimes, namely $v_0\tau/S \sim 1$ and $v_0\tau/S \geq 1$, respectively. Colored symbols (and dotted lines drawn as a guide to the eye) are obtained from numerical simulations while solid colored lines show the theoretical prediction (11) (in the regions where $\Lambda(x)$, given by Eq. (19), is positive definite). Simulations are realized in one spatial dimension with $\tau k/\gamma = 1$ and $\alpha = 0.9$.

When $v_0\tau/S$ increases, e.g., due to a shorter periodicity $S$ of the swim-velocity $u(x)$, $\langle v^2(x)\rangle$ decreases upon moving away from the potential minimum. This is consistent with the scenario observed in Fig. 4 (a): the particles accumulate in the regions where they move slowly and the velocity variance is small. Such a result agrees with the observed behavior in the potential-free case (such that $u(x)$ coincides the particle velocity), where the particles accumulate in regions corresponding to the minima of $u(x)$, according to the law $\rho(x) \sim 1/u(x)$. In this regime, the comparison between numerical data and the theoretical prediction (11) (with $\Lambda(x)$ given by Eq. (19)) shows a good agreement.

For larger values of $v_0\tau/S$ (large-persistence regime), the velocity variance shows a more complex profile (see Fig. 4 (d)), which resembles the oscillating shape of $u(x)$. In particular, $\langle v^2(x)\rangle$ is very small near the peaks of the density distribution, while it assumes larger values in the regions where the density is very small and the probability of finding a particle is very low. This finding is consistent with the fact that active particles accumulate in the regions where the velocity variance is small and the observation of an increasing number of such regions for increasing $v_0\tau/S$. Finally, in this regime, the prediction (11) reproduces quite well the behavior near the origin but fails further away from it, specifically, in the regions where the effective friction displays negative values, $\Lambda(x) < 0$.

# 5   Conclusion

In this paper, we have investigated the stationary behavior of an active particle subject to two competing spatially dependent drivings: the self-propulsion velocity and the external force. While the two mechanisms were already investigated separately, to the best of our knowledge, this is the first time that their interplay has been considered. Starting from a Fokker-Planck description of the particle's dynamics in our generalized AOUP model [74], we have developed a theoretical treatment, which provides the steady-state distribution (9) of both positions and velocities as a function of the input potential and of the swim-velocity profile. The theory presented here contains as special cases both the UCNA describing the time evolution of distribution of positions and velocities of an AOUP with constant swim velocity in an external field [59, 88], and the recent theory of a free AOUP driven by an inhomogeneous propulsion force, which is exact in the stationary case [74]. Our theoretical method is exact in the small-persistence regime, where it is consistent with the results obtained through an exact perturbative method, and also provides a useful approximation to qualitatively predict the shape of the distributions in the large-persistence regime.

Specifically, we have applied our theory to a one-dimensional AOUP in a sinusoidal motility landscape subject to a harmonic potential, observing intriguing effects, which arise from the interplay between these two fields. Indeed, it is known that an AOUP with constant swim velocity in a harmonic potential is described by Gaussian distributions for position and velocity [60, 64] while the velocity distribution of a potential-free AOUP in a motility landscape is described by a Gaussian velocity distribution [74]. The joint effect of harmonic confinement and motility landscape revealed an intriguing scenario determined by the joint action of the self-propulsion velocity gradient and the external force. While, in the regime of small persistence, both the density and the velocity distributions are bell-shaped and well-approximated by Gaussians, we predict that, as the persistence length becomes comparable with the spatial period of the swim velocity, a transition from a unimodal to a bimodal density occurs, also accompanied by strong non-Gaussian effects in the velocity distribution. Interestingly, in the large-persistence regime, as the density shows multi-modality, the velocity distribution becomes bimodal in the spatial regions between two successive peaks of the density.

Despite our particular attention to one spatial dimension, we recall that we practically obtain the same results for a planar geometry in higher spatial dimensions and that we expect that our theory is also suitable for sufficiently well-behaved potentials and velocity fields in other geometries (compare the discussion in Sec. 3.1). While, for active colloids, the emergence of an additional effective torque due to the spatial modulation of the swim velocity could be responsible for an even more complex phenomenology [16, 19], we outline that our theory should be suitable in the case of engineered bacteria whose velocity profile can be manipulated by external light [25, 27]. From a pure theoretical perspective, our techniques may also be extended and applied to more complex dynamics, for instance accounting for the presence of thermal noise [73, 75], a spatially dependent torque [16, 19], or additional competing nonconservative force fields like a Lorentz force [102]. A final challenging research point concerns the dynamical properties of our model and, in particular, the extension of the theory to time-dependent swim-velocity profiles $u(x, t)$, for instance in the form of traveling waves [23, 24, 49].

In conclusion, we have shown that the interplay between an external force and a spatially modulating swim velocity can be used to tune the behavior of a confined active particle, for instance by locally increasing the kinetic temperature or by forcing the particles to accumulate in particular spatial regions with different probability. We demonstrated that by combining these two physically distinct effects, it is possible to generate complex density patterns through relatively simple fields, as in our example a harmonic trap and a periodic velocity landscape.

In practice, realizing such particle distributions through a single external field is surely more involved due to the required complex form of the potential. The possibility to fine-tune the stationary properties of active particles in experimental systems by adapting both the external force and the swim velocity opens up a new avenue for future applications and developments.

## Acknowledgements

We gratefully acknowledge Michael Klatt, Ivo Buttinoni and Claudio Maggi for sharing valuable insights into the mathematical and experimental relevance of considering the interplay of spatially dependent self propulsion and external confinement. LC and UMBM warmly thank Andrea Puglisi for letting us use the computer facilities of his group and for discussions regarding some aspects of this research.

**Funding information** LC and UMBM acknowledge support from the MIUR PRIN 2017 project 201798CZLJ. LC acknowledges support from the Alexander Von Humboldt foundation. RW and HL acknowledge support by the Deutsche Forschungsgemeinschaft (DFG) through the SPP 2265, under grant numbers WI 5527/1-1 (RW) and LO 418/25-1 (HL).

## A   Derivation of the auxiliary dynamics (6)

To derive the auxiliary dynamics (6), we start from Eq. (2a) choosing $D_\text{t} = 0$. We recall that following Ref. [75] it is possible to generalize the procedure also to include the more general case with $D_\text{t} > 0$. At first, we take the time-derivative of Eq. (2a), obtaining:

$$\gamma \ddot{\mathbf{x}} = -\nabla\nabla U \cdot \dot{\mathbf{x}} + \gamma \boldsymbol{\eta} \left( \frac{\partial}{\partial t} + \mathbf{v} \cdot \nabla \right) u(\mathbf{x}, t) + \gamma u(\mathbf{x}, t) \dot{\boldsymbol{\eta}}. \tag{A.1}$$

By defining the $\mathbf{v} = \dot{\mathbf{x}}$ as the particle velocity and replacing $\dot{\mathbf{f}}_\text{a}$ with the dynamics (2b), we get:

$$\gamma \dot{\mathbf{v}} = -\nabla\nabla U \cdot \mathbf{v} + \gamma \boldsymbol{\eta} \left( \frac{\partial}{\partial t} + \mathbf{v} \cdot \nabla \right) u(\mathbf{x}, t) + \gamma u(\mathbf{x}, t) \left( -\frac{\boldsymbol{\eta}}{\tau} + \frac{\sqrt{2}}{\sqrt{\tau}} \boldsymbol{\chi} \right). \tag{A.2}$$

Finally, by replacing $\boldsymbol{\eta}$ in favor of $\mathbf{v}$ and $\mathbf{x}$, taking advantage of the relation (2a), we obtain the dynamics (6).

We remind that this sequence of operation is fully equivalent to performing a change of variables, by considering that the dynamics (2a) is a deterministic relation that allows us to replace $\boldsymbol{\eta}$ with $\mathbf{x}$ and $\mathbf{v}$. The Jacobian matrix $\mathcal{J}$ of this transformation $\mathbf{a} = (\mathbf{x}, \boldsymbol{\eta}) \to \mathbf{b} = (\mathbf{x}', \mathbf{v})$ with $\mathbf{x}' = \mathbf{x}$ reads:

$$\mathcal{J} = \frac{\partial \mathbf{b}}{\partial \mathbf{a}} = \begin{bmatrix} \frac{\partial \mathbf{x}'}{\partial \mathbf{x}} & \frac{\partial \mathbf{x}'}{\partial \boldsymbol{\eta}} \\ \frac{\partial \mathbf{v}}{\partial \mathbf{x}} & \frac{\partial \mathbf{v}}{\partial \boldsymbol{\eta}} \end{bmatrix} = \begin{bmatrix} \mathbf{I} & \mathbf{0} \\ \mathbf{0} & \mathbf{I}u(\mathbf{x}) \end{bmatrix}. \tag{A.3}$$

The determinant of this matrix, $\det[\mathcal{J}] = u(\mathbf{x})$, yields the Jacobian of the transformation as stated in Eq. (4).

## B   Derivation of predictions (9) and (10)

To predict the shape of the stationary probability distributions, $p(\mathbf{x}, \mathbf{v})$ and $\rho(\mathbf{x})$, stated in Sec. 3, we start from the dynamics in the variables $\mathbf{x}$ and $\mathbf{v}$, namely Eq. (6), for a static profile

of the swim velocity, $u(\mathbf{x})$. Switching to the Fokker-Planck equation for $p = p(\mathbf{x}, \mathbf{v}, t)$, we obtain the exact time evolution:

$$\partial_t p = \nabla_v \cdot \left( \frac{\mathbf{\Gamma}}{\tau} \cdot \mathbf{v}p + \frac{u^2(\mathbf{x})}{\tau} \nabla_v p \right) - \mathbf{v} \cdot \nabla p + \nabla \cdot \frac{\nabla U}{\gamma \tau} p - \nabla_v \cdot \frac{[\gamma \mathbf{v} + \nabla U]}{\gamma u(\mathbf{x})} (\mathbf{v} \cdot \nabla) u(\mathbf{x}), \qquad \text{(B.1)}$$

where $\nabla$ and $\nabla_v$ are the vectorial derivative operators in position and velocity space, respectively. Balancing the diffusion term (proportional to the Laplacian of $\mathbf{v}$) and the other effective friction terms (say the one linearly proportional to $\mathbf{v}$), we get the approximate condition [81]:

$$0 = \nabla_v \cdot \left( \frac{\mathbf{\Lambda}}{\tau} \cdot \mathbf{v}p + \frac{u^2(\mathbf{x})}{\tau} \nabla_v p \right), \qquad \text{(B.2)}$$

with the effective friction matrix

$$\mathbf{\Lambda}(\mathbf{x}) = \mathbf{I} + \frac{\tau}{\gamma} \nabla \nabla U(\mathbf{x}) - \frac{\tau}{\gamma} \nabla U(\mathbf{x}) \frac{\nabla u(\mathbf{x})}{u(\mathbf{x})}, \qquad \text{(B.3)}$$

that has been defined in Eq. (8). The condition (B.2) corresponds to requiring that the divergence of the irreversible (with respect to time-reversal transformations) currents is zero. To proceed further, we require that the irreversible currents vanish, i.e., that the expression in the brackets of Eq (B.2) is zero in the same spirit of Ref. [88]. This choice is consistent with an effective equilibrium approach and allows us to find the explicit approximate steady-state solution for $p(\mathbf{x}, \mathbf{v})$ as

$$p(\mathbf{x}, \mathbf{v}) \propto g(\mathbf{x}) \exp \left( -\frac{\mathbf{v} \cdot \mathbf{\Lambda}(\mathbf{x}) \cdot \mathbf{v}}{2u^2(\mathbf{x})} \right), \qquad \text{(B.4)}$$

where $g(\mathbf{x})$ is a function purely depending on $\mathbf{x}$, which is still to be determined. Expressing $g(\mathbf{x}) = \rho(\mathbf{x}) \sqrt{\det[\mathbf{\Lambda}(\mathbf{x})]}/(\sqrt{2\pi}u(\mathbf{x}))$ without loss of generality, we obtain Eq. (9), where $\rho(\mathbf{x})$ represents the density of the system derived below.

To determine the function $\rho(\mathbf{x})$, we first identify the acceleration term [77]

$$u(\mathbf{x}) \frac{d}{dt} \frac{\mathbf{v}}{u(\mathbf{x})} = \dot{\mathbf{v}} - \frac{\mathbf{v}}{u(\mathbf{x})} (\mathbf{v} \cdot \nabla) u(\mathbf{x}) \qquad \text{(B.5)}$$

in the dynamics (6) with $\partial u(\mathbf{x})/\partial t = 0$, see appendix C for more details. Assuming that the velocity relaxes faster than the position (as for example in the small-persistence regime) allows us to neglect both these terms in Eq. (6), obtaining the following overdamped equation:

$$\dot{\mathbf{x}} = -\frac{1}{\gamma} \mathbf{\Lambda}^{-1} \cdot \nabla U + \sqrt{2\tau} u(\mathbf{x}) \mathbf{\Lambda}^{-1} \cdot \boldsymbol{\chi}. \qquad \text{(B.6)}$$

From this dynamics, it is convenient to switch to the effective Smoluchowski equation for the density of the system, $\rho(\mathbf{x}, t)$, and use the Stratonovich convention, obtaining:

$$\frac{\partial \rho}{\partial t} = \frac{\partial}{\partial x_i} \left( \frac{1}{\gamma} \Lambda_{ij}^{-1} \left( \frac{\partial U}{\partial x_j} \right) \rho + \tau u \Lambda_{ik}^{-1} \frac{\partial}{\partial x_j} \left[ \Lambda_{jk}^{-1} u \rho \right] \right). \qquad \text{(B.7)}$$

Here and in what follows, we have explicitly written Latin indices for the spatial components of vectors and matrices and adopted also the Einstein's convention for repeated indices, for convenience.

To proceed, we assume the zero-current condition (as in Refs. [59, 103]), obtaining an effective equation for the stationary density $\rho(\mathbf{x})$:

$$\frac{1}{\gamma} \Lambda_{ij}^{-1} \left( \frac{\partial U}{\partial x_j} \right) \rho + \tau u \Lambda_{ik}^{-1} \frac{\partial}{\partial x_j} \left[ \Lambda_{jk}^{-1} u \rho \right] = 0. \qquad \text{(B.8)}$$

Multiplying by $\Lambda_{lh}\Lambda_{hi}$ and summing over repeated indices, we get the following relation after some algebraic manipulations

$$\frac{1}{\gamma\tau u^2}\Lambda_{lj}\frac{\partial U}{\partial x_j} + \Lambda_{lk}\frac{\partial}{\partial x_j}\Lambda_{jk}^{-1} + \frac{\Lambda_{lk}\Lambda_{jk}^{-1}}{u\rho}\frac{\partial}{\partial x_j}[u\rho] = 0, \tag{B.9}$$

whose solution for the density distribution $\rho(\mathbf{x})$ reads:

$$\rho(\mathbf{x}) \approx \frac{\mathcal{N}}{u(\mathbf{x})}\exp\left(\frac{1}{\gamma\tau}\int^{\mathbf{x}}d\mathbf{y}\cdot\frac{\mathbf{\Lambda}(\mathbf{y})\cdot\mathbf{F}(\mathbf{y})}{u^2(\mathbf{y})} + \int^{\mathbf{x}}d\mathbf{y}\cdot\mathbf{\Lambda}(\mathbf{y})\cdot\nabla\cdot\mathbf{\Lambda}^{-1}(\mathbf{y})\right). \tag{B.10}$$

Finally, by assuming a planar symmetry for both $u$ and $U$, we have $\nabla\cdot\mathbf{\Lambda}^{-1} \equiv \hat{e}_x\cdot\partial\mathbf{\Lambda}^{-1}/\partial x$, where $\hat{e}_x$ denotes the unit vector corresponding to the coordinate $x$, and can therefore use the explicit Jacobi relation

$$\mathbf{\Lambda}\cdot\hat{e}_x\cdot\frac{\partial\mathbf{\Lambda}^{-1}}{\partial x} = -\frac{1}{\det[\mathbf{\Lambda}]}\hat{e}_x\frac{\partial\det[\mathbf{\Lambda}]}{\partial x} = -\hat{e}_x\frac{\partial\ln\det[\mathbf{\Lambda}]}{\partial x}, \tag{B.11}$$

for the determinant $\det\mathbf{\Lambda}$ of a matrix $\mathbf{\Lambda}$. We remark that the general relation

$$\mathbf{\Lambda}\cdot\nabla\cdot\mathbf{\Lambda}^{-1} = -\nabla\ln\det[\mathbf{\Lambda}] \tag{B.12}$$

only holds in the above planar case (B.11) or for a constant swim velocity $u(\boldsymbol{x}) = v_0$, see also appendix B of Ref. [73]. However, since there are no conceptual differences, we can plug the approximation (B.12) into the prediction (B.10) to obtain the compact representation (10) of $\rho(\boldsymbol{x})$ in the main text.

The same stationary condition (B.9) can be obtained using the Fox approach [80] (when generalized to multiple components [104,105]), while the corresponding time evolution differs from the UCNA dynamics (B.7) by the additional occurrence of the factors $\Lambda_{ij}^{-1}$ and $\Lambda_{ik}^{-1}$ therein. Note that, if we do not neglect the thermal Brownian noise in Eq. (2), also the stationary predictions of Fox and UCNA differ, even for a spatially constant swim velocity [73].

# C  Effective change of variables for multiplicative noise

In this appendix, we briefly discuss a change of variables [77,83] which allows us to formally absorb the effect of the spatially dependent swim velocity into an effective force. Since $u(\mathbf{x})$ is positive definite, we can define a new variable $\tilde{\mathbf{x}}$ such that

$$\dot{\tilde{\mathbf{x}}} := \frac{\dot{\mathbf{x}}}{u(\mathbf{x})}. \tag{C.1}$$

Substituting into Eq. (3) yields

$$\dot{\tilde{\mathbf{x}}} = \frac{\gamma^{-1}\mathbf{F}(G^{-1}(\tilde{\mathbf{x}}))}{u(G^{-1}(\tilde{\mathbf{x}}))} + \boldsymbol{\eta}, \tag{C.2}$$

where $G^{-1}(\tilde{\mathbf{x}})$ denotes the inverse of the bijection $G(\mathbf{x}) = \tilde{\mathbf{x}}$ mediating the variable transform $\mathbf{x}\to\tilde{\mathbf{x}}$.

There are two possible ways of interpreting Eq. (C.2). First, on the right-hand side, we can identify an effective external force and an additive noise. It is thus possible to argue that the multiplicative nature of the swim-velocity field $u(\mathbf{x})$ can be absorbed into a convoluted expression of an external potential (and implicitly in the definition of $\tilde{\mathbf{x}}$). However, since the

new coordinate $\tilde{\mathbf{x}}$ has the dimension of time rather than length, this is only a mathematical analogy, which, owing to the simpler form might be useful to find an explicit solution as, for example, in the case of a Run&Tumble model in one dimension [83]. Transforming back to the original coordinates explicitly requires the notion of the swim velocity $u(\mathbf{x})$, compare Eq. (C.1). Second, multiplying both sides of Eq. (C.2) with $u(\mathbf{x})$, we recover an equation with a multiplicative noise in which the combined term $u(\mathbf{x})\dot{\tilde{\mathbf{x}}}$ has a proper dimension of a velocity. Moreover, upon first calculating the second time derivative $\ddot{\tilde{\mathbf{x}}}$ and then multiplying with $u(\mathbf{x})$, we are able to identify the acceleration term which can be neglected to obtain the UCNA [77], compare Eq. (B.5). In other words, the physical interpretation of our results is the same whether or not we perform such a change of variables.

# D  Multi-scale technique: derivation of Eq. (14)

In this appendix, we derive the perturbative result (14) for the probability distribution $p(x, v)$ in the one-dimensional active system described by the Fokker-Planck equation (13). We adopt the multiple-time-scale technique, which is designed to deal with problems with fast and slow degrees of freedom. In the regime of small persistence time (where $\tau$ is the smallest time scale of the system), the dynamics (13) exhibits the separation of time scales: in this case, the particle velocity rapidly arranges according to its stationary distribution and the spatial distribution evolves on a slower time scale.

To derive the multiple-time expansion, let us introduce the following dimensionless variables:

$$\tilde{t} = t\frac{v_0}{S}, \tag{D.1}$$

$$X = \frac{x}{S}, \tag{D.2}$$

$$V = \frac{v}{v_0}, \tag{D.3}$$

$$\tilde{F}(X) = -\frac{S}{v_0^2 \tau \gamma}\frac{\partial U(x)}{\partial x}, \tag{D.4}$$

$$\tilde{\Gamma}(X) = 1 - \tau^2 \frac{v_0^2}{S^2}\frac{\partial \tilde{F}(X)}{\partial X}, \tag{D.5}$$

$$w(X) = \frac{u(x)}{v_0}, \tag{D.6}$$

and the small expansion parameter $\zeta^{-1} \propto \tau$, where

$$\zeta = \frac{S}{\tau v_0} \tag{D.7}$$

is the ratio between the spatial period of the swim velocity $S$ and the persistence length of the self-propulsion velocity $v_0\tau$. With our choice, a large (small) value of $\zeta$ corresponds to the small-persistence (large-persistence) regime. Now, we express the Fokker-Planck equation (13) in these variables and find:

$$\frac{\partial P(X, V, \tilde{t})}{\partial \tilde{t}} + V\frac{\partial}{\partial X}P + \tilde{F}(X)\frac{\partial}{\partial V}P + \frac{1}{\zeta}R(X)\frac{\partial}{\partial V}VP + \frac{1}{w(X)}\frac{\partial}{\partial X}w(X)\frac{\partial}{\partial V}(V^2 P) = \zeta L_{\text{FP}}P, \tag{D.8}$$

where we have further introduced the operator

$$L_{\text{FP}} \equiv \frac{\partial}{\partial V}\left(V + w^2(X)\frac{\partial}{\partial V}\right), \tag{D.9}$$

and the function

$$R(X) \equiv \left[ \frac{\partial \tilde{F}}{\partial X} - \tilde{F}(X) \frac{1}{w(X)} \frac{\partial}{\partial X} w(X) \right], \tag{D.10}$$

for convenience.

To develop our perturbative solution, we notice that the local operator $L_{\mathrm{FP}}$ is proportional to the inverse expansion parameter $\zeta$ in Eq. (D.8). We find that $L_{\mathrm{FP}}$ has the following integer eigenvalues:

$$\nu = 0, -1, -2, \ldots \tag{D.11}$$

and the Hermite polynomials as eigenfunctions:

$$H_\nu(X,V) = \frac{(-1)^\nu}{\sqrt{2\pi}} (w(X))^{(\nu-1)} \frac{\partial^\nu}{\partial V^\nu} e^{-\frac{V^2}{2w^2(X)}}. \tag{D.12}$$

Using these basis functions, we obtain the ansatz to write the solution of the partial differential equation as a linear combination:

$$P(X,V,\tilde{t}) = \sum_{\nu=0}^{\infty} \phi_\nu(X,\tilde{t}) H_\nu(X,V). \tag{D.13}$$

Upon substituting the expansion (D.13) in Eq. (D.8) and replacing $L_{\mathrm{FP}}$ by its eigenvalues, we obtain the equation:

$$\begin{aligned} -\zeta \sum_\nu \nu \phi_\nu H_\nu =& \sum_\nu \frac{\partial \phi_\nu}{\partial \tilde{t}} H_\nu \\ &+ \sum_\nu V H_\nu \frac{\partial}{\partial X} \phi_\nu + \sum_\nu \phi_\nu V \frac{\partial}{\partial X} H_\nu + \tilde{F} \sum_\nu \phi_\nu \frac{\partial}{\partial V} H_\nu \\ &+ \frac{w'}{w} \sum_\nu \phi_\nu \frac{\partial}{\partial V} V^2 H_\nu + \frac{1}{\zeta} R \sum_\nu \phi_\nu \frac{\partial}{\partial V} V H_\nu, \end{aligned} \tag{D.14}$$

from which we must determine the unknown functions $\phi_\nu(x,t)$.

Now, instead of truncating arbitrarily the infinite series in Eq. (D.14) at some order $\nu$, we consider the multiple-time expansion which orders the series in powers of the small parameter $1/\zeta$. In such a way we perform an expansion near the equilibrium solution. To this end, each amplitude $\phi_\nu$ (apart from $\phi_0(X,\tilde{t})$ which is of order $\zeta^0$) is expanded in powers of $1/\zeta$ as:

$$\phi_\nu(X,\tilde{t}) = \sum_{s=0}^{\infty} \frac{1}{\zeta^s} \psi_{s\nu}(X,\tilde{t}). \tag{D.15}$$

Then, we replace the actual probability distribution $P(X,V,\bar{t})$ by an auxiliary distribution $P_{\mathrm{a}}(X,V,\bar{t}_0,\bar{t}_1,\bar{t}_2,\ldots)$, which reads:

$$P_{\mathrm{a}} = \sum_{s=0}^{\infty} \frac{1}{\zeta^s} \sum_{\nu=0}^{\infty} \psi_{s\nu}(X,\bar{t}_0,\bar{t}_1,\bar{t}_2,\ldots) H_\nu(X,V). \tag{D.16}$$

This distribution depends on many time variables $\{\bar{t}_s\}$, associated with the perturbation order $s$, which are defined as $\bar{t}_s = \bar{t}/\zeta^s$. The time derivative with respect to $\bar{t}$ is then expressed as the sum of partial time-like derivatives:

$$\frac{\partial}{\partial \bar{t}} = \frac{\partial}{\partial \bar{t}_0} + \frac{1}{\zeta} \frac{\partial}{\partial \bar{t}_1} + \frac{1}{\zeta^2} \frac{\partial}{\partial \bar{t}_2} + \ldots. \tag{D.17}$$

Substituting the expansions (D.15) and (D.17) into Eq. (D.14) one obtains at each order $1/\zeta^s$ and for each Hermite function an equation involving the amplitudes $\psi_{s\nu}(X, \tilde{t})$. The perturbative structure of the resulting set of equations is such that the amplitudes $\psi_{s\nu}(X, \tilde{t})$ can be obtained by the amplitudes of the lower order $(s-1)$. In particular, we find the following equation for $\psi_{00} = \phi_0$

$$\frac{\partial \psi_{00}(X, \bar{t})}{\partial \tilde{t}} = \frac{1}{\zeta}\frac{\partial}{\partial X}\Big[w(X)\frac{\partial}{\partial X}\big(w(X)\psi_{00}\big) - \tilde{F}(X)\psi_{00} + \frac{1}{\zeta^2}w(X)\frac{\partial}{\partial X}\big(w(X)R\psi_{00}\big)\Big], \quad \text{(D.18)}$$

whose steady-state solution reads

$$\psi_{00}(X) = \frac{\mathcal{N}}{w(X)}\Big(1 - \frac{1}{\zeta^2}R(X)\Big) \times \exp\left[\int^X dy \frac{\tilde{F}(y)}{w^2(y)}\Big(1 - \frac{1}{\zeta^2}R(y)\Big)\right], \quad \text{(D.19)}$$

where $\mathcal{N}$ is a normalization factor. In our perturbative procedure, all the remaining amplitudes are expressed in terms of the pivot function $\psi_{00}(X)$. The steady-state amplitudes of the higher-order Hermite polynomials are given by:

$$\psi_{22} = \frac{1}{2}R(X)\psi_{00}, \quad \text{(D.20)}$$

$$\psi_{33}(X) = -\frac{1}{6}w(X)\psi_{00}(X)\frac{\partial}{\partial X}R(X), \quad \text{(D.21)}$$

$$\psi_{42} = -\frac{3}{2}\frac{\partial}{\partial X}[w(X)\psi_{33}] + R(X)\psi_{22}, \quad \text{(D.22)}$$

$$\psi_{44}(X) = -\frac{1}{4}\Big(\frac{\partial}{\partial X}[w(X)\psi_{33}] - R(X)\psi_{22}\Big), \quad \text{(D.23)}$$

where we have reported only the nonvanishing coefficients for $s \leq 4$. Note that, if $\nu > s$, the coefficients $\psi_{s\nu}$ are always zero.

Once Eq. (D.19) and the coefficients of the double series (D.16) have been determined, one returns to the original dimensional variables and obtains the perturbative result for $p(x, v)$ reported in Eq. (14).

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
