# Peer review of "Active particles driven by competing spatially dependent self-propulsion and external force"

_SciPost Physics, doi:SciPost Phys. 13, 065 (2022)_

## Round 1 · Referee Report · Anonymous (Referee 1) · 2022-3-31

Strengths

1 - The authors provide a comprehensive study of their model, both from the theoretical and numerical points of view.
2 - References to previous literature on the subject are numerous and relevant.
3 - The authors give clear interpretations of their results, supplemented by general qualitative arguments.

Weaknesses

1 - Approximations made to derive the theoretical predictions are not motivated enough in the main text.

Report

The authors study an AOUP in a confining harmonic potential with a velocity profile varying periodically in space. It is already known that active particles accumulate where they move slower, this study shows that the competition with an additional confining potential creates unfavourable regions of space — where the restoring force overcomes the propulsion force — such that the density profile transitions from a normal distribution at low persistence length to a multimodal distribution at large persistence. Numerical results are carefully confronted to theoretical expectations

The paper provides, for this simple model, a detailed account of the physics at play in the different regimes of persistence, and should thus serve as a solid basis for follow-up studies on the control of active matter. As such, I recommend publication to SciPost Physics.

Requested changes

1 - The limits of validity for the unified coloured noise approximation (UCNA) are not detailed in the main text. To which quantities should the persistence time be compared in order to characterise the small/large-persistence regimes? When should we expect this approximation to fail?
2 - How strong is the approximation of vanishing probability current? What are the consequences of such an approximation?

Minor comments:
3 - Above equation (21), is the second derivative of the density profile taken at x=0?
4 - On top of page 12, (3) is not a prediction, should it rather be (9)?
5 - In Fig. 4, could you add density profiles so that comparisons can be made similarly to all other figures?

---

## Round 1 · Referee Report · Anonymous (Referee 2) · 2022-4-13

# Active particles driven by competing spatially dependent self-propulsion and external force

This paper studies an active particle (AOUP) with spatially-dependent activity in a confining potential and more especially in a harmonic well. The authors have a written english of good quality which makes the reading pleasant. In the first half of the paper, they tackle their model analytically by using two approaches: the UCNA approximation and an exact perturbative scheme. While I did not try to reproduce their computations in details, their study seems rigorous and well-grounded in the literature: the uncontrolled assumptions are stated and the terms neglected are mentioned. In the second half of the paper, the authors perform numerical work to support their study. They distinguish three regimes for the density depending on the parameter $v_0\tau/S$: monomodality, bimodality and multimodality. For each of these three regimes, they also supply the distribution of the velocity as well as its variance when the spatial position is varied. Despite all these results, I had troubles to understand the novelty or the take-home message of the paper. In particular, I have four major concerns which I develop below

- To which extent is taking a spatially-dependent self-propulsion $u(x,t)$ different than just having a single AOUP in a complicated potential? From the authors' results, it does not seem that there is unexpected physics due to the spatially-dependent $u(x,t)$. In particular, all the features highlighted by the authors for an AOUP with spatially-dependent self-propulsion (accumulation near walls for example) are already present for an AOUP with constant self-propulsion. I have remarked that, when $u(x)$ is a bijection, one can map the dynamics (3) of the authors to the dynamics of an AOUP with spatially-independent self-propulsion by making the change of variable $\tilde{u} = \ln(u(x))$. Indeed, dividing both sides of (3) by $u(x)$ leads to

$$\frac{\dot{x}}{u} = -\frac{\nabla U}{u} + \eta \qquad \Leftrightarrow \qquad \dot{\tilde{u}} = g(\tilde{u}) + \eta \qquad (1)$$

where $g(\tilde{u}) = \nabla U(u^{-1}(e^{\tilde{u}}))/e^{\tilde{u}}$ and $\eta$ is an Orstein-Uhlenbeck process as described in (2b) of the paper. From (1), I intuitively do not expect much differences between an AOUP with a spatially-dependent self-propulsion and one with a constant self-propulsion in a more complicated potential. Have the authors studied the question?

- The authors' results can be obtained by applying two main results of active matter that are already well-established. The first one is that active particle with spatially-dependent

self-propulsion $u(x)$ accumulate where they are slow as $\rho(x) \sim 1/u(x)$ (Refs [28] and [29] in the introduction). The second one is that confining an active particle with a potential $U$ leads to an accumulation at the position where the force $\nabla U(x)$ is equal to the self-propulsion amplitude $u$ (Ref [71] and [86] from the authors). This last result, which has been documented, could also be seen as a consequence of the former one: the confining force effectively reduces the self-propulsion of the active particle which in turn triggers an accumulation due to slowness.

Figure 1 and figure 2 are intuitively explained in light of these two established results. For example, in figure 1d, spikes of the density correspond to positions where $\nabla U(x) = u(x)$. In figure 2d, the two spikes close to the center correspond to local minima of $u(x)$ while the two highest peaks correspond to both local minima of $u(x)$ and positions where $\nabla U(x) = u(x)$. Thus, it is not surprising that these spikes are the highest ones: the effective self-propulsion at these positions is lowered by the potential compared to the self-propulsion at the center.

Finally, I believe that the bimodality of the velocity distribution displayed in figure 3h is not surprising; it is due to particles coming from the cluster on the right and going leftway as well as to particles coming from the cluster on the left and going rightway. Because the confining force increases away from the center, the position of the spike with negative velocity diminushes from yellow to brown while the trend is opposite for the spike with positive velocity.

- I do not see what is the new physics brought forward in this work which has not already been discussed in Ref [70] by the authors. For example, in the present paper, the authors affirm that "the interplay between the external force and the modulation of the swim velocity can be used to manipulate the behavior of a confined active particle, for instance by locally increasing the kinetic temperature or by forcing the particles to accumulate in distinct spatial regions with different probability". However, in their previous work [70], the authors have already demonstrated that the density of their model follows the law $\rho(x) \sim 1/u(x)$ in the absence of potential. Thus there is no need of a confining potential if one wants to sort and accumulate active particles in distinct regions: a spatially-dependent self-propulsion is enough. What is the motivation and the new physics when studying the interplay between a potential and a spatially-dependent self-propulsion?

- As the authors point out in their work [70], there is already a literature studying spatially-dependent self-propulsion amplitude and memory time for AOUP (ref [61] for example).

In [70], the authors detailed, from a mathematical point of view, why their model (2) is actually different from the model studied in [61]. However, from a physical point of view, the difference remains unclear: what are the physical features present in (2) that are not in [61]? In particular, when it comes to experiments [25-27], what is the correct model one should use? It would be interesting if the authors show that their model is indeed the one relevant for the experiments.

Given the four major points discussed above, I would not recommend this paper for publication in SciPost. I believe that the authors should more significantly ground the novelty as well as the physical motivation that drove the writing of this paper. Beside the important points raised above, I only have minor remarks which I listed below:

- I did not see an indication about the dimension considered in simulations: are there performed in $d = 1$ or $d = 2$?

- the change of variable presented right after (3) might benefit from a clearer presentation. I was first confused about whether $v$ included the confining force.

- In the appendix eq (27), one side has $u(x)$ while the other has $u(x, t)$

- Right before the part on density distribution: "when $v_0\tau/S$ grows, the spatial period of u(x) increases" should be "when $v_0\tau/S$ grows, the spatial period of u(x) decreases" if I understood correctly the authors ($S$ being the spatial period).

- Right in the beginning of the part Velocity distribution, "display an almost Gaussian shape in agreement with Eq (3)" should probably be "display an almost Gaussian shape in agreement with Eq (10)". The same reference lapsus is also found later in the same part "Gaussian distribution with space-dependent variance given by Eq (3)".

- On figure 3; is the x-axis rescaled? Is it $x$ or $\tilde{x}$?

- In the end of the part called profile of the kinetic temperature, I could not understand well the phrase "the variance of the particle velocity becomes steeper and decreases to zero in the regions which are not explored by the particle. This is consistent with the scenario observed in Fig. 1 (c): the particles accumulate in the regions where they move slowly and the velocity variance is small". How can particles accumulate in regions that they do not explore?

---

## Round 1 · Referee Report · Anonymous (Referee 3) · 2022-5-5

Report

The authors propose a lecture note about Active Ornstein-Uhlenbeck particles in a spatially-dependent motility landscape. Working in the Unified Colored Noise Approximation, they provide an approximate stationary probability distribution and then they check its validity in the specific case of a 1-dimensional system embedded into a harmonic trap and whose motility is modulated by a sinusoidal function. I found the subject of the lecture note interesting and the problem they focused on looks timely. I shall support for publication once the authors address satisfactorily all my comments listed below.

Comments 1. It is not completely clear to me how the velocity of the particle is controlled in this model. In the sense that $\eta$ fluctuates with a variance that is proportional to $1/\tau$ and then the velocity is basically $u(x,t) \eta$. For $u(x,t)=v_0$, it looks to me that velocity is not really controlled unless you change the variance of the OU process. I might be wrong, however, I think that a general reader might be getting wrong like me and thus I would suggest the authors clarify this point.

  1. I did not find clear the presentation in the section “Velocity description of AOUP”. It seems to me that they just perform the time derivative of Eq. (3) and then they consider the usual UCN approximation ($\tau \ddot{x}=0$). I do not think an appendix of just two lines is required (see eq. (22) and (23)) since this derivation might be very useful for the general reader. Moreover, they introduce the Jacobian but a) they do not write its expression, and b) Does occur any problem if J=0? I am asking that because after (1) they require $u\geq0$, however, $|J|=u$ and the approximated solution seems to require $u>0$. Is this a problem of the approximation or, more in general, the model is not defined for $u=0$?

  2. I think Eq. (7) requires some warnings because, in the region where the potential develops negative curvatures, one ends with negative friction. I understand the authors choose a safe potential with always positive curvature (basically this is also the reason why for large $\tau$ one can obtain a good approximation of stationary configurations). For instance, it looks to me that even in a small $\tau$ limit negative and large curvatures might cause problems. (a) Could the authors comment on that and also include relevant references? (b) I think a similar discussion is required once they introduce $\Lambda(x)$

  3. In section “Theoretical Predictions” should contain all the relevant information about the approximations made to arrive at Eqs. (9) and (10) are valid. Moreover, in Eq. (10), obtained within UCN, there is a $\det \Lambda(x)$ that does not appear in Fox: this is because Fox is a small $\tau$ approximation (the authors write “The same $\rho(x)$ can be obtained the path-integral method proposed by Fox”). I understand the small $\tau$ of UCN brings to Fox, however, in UCN one just asks for large friction. (a) Could the authors clarify this point? Also, (10) has clearly a problem with negative values of $\Lambda$ but also with$ u=0$, (b) could the authors provide a list of conditions under which the approximated solution is valid? (c) Again, I think there are conditions for writing Eq. (9). In appendix (B) they show how to obtain rho(x) but I did not find information in Eq. (9).

  4. The theory has been checked against numerical simulations in one spatial dimension. Does the approximated solution presented in (10) hold also in two dimensions? I think the authors should clarify the reasons why they focus their attention on one-dimensional problems.

  5. In the section “The harmonic oscillator” they write “we have shown that our analytical predictions from Eqs. (9), (10) and (11) are exact in the small persistence regime through analytical arguments…”. I do not see where a small tau approximation enters (10).

  6. I think it might be very useful to the reader if the authors could add more details about the comparison between numerical results with Eq. (10). (a) Could the authors write the expression they use in 1 dim? (b) Could the authors say how the comparison has been performed, i.e., they integrate numerically eq. (9)?

  7. In “Conclusions” they write “we have developed a theoretical treatment, applicable to rather general choices of confining potentials and inhomogeneous swim velocities…”. I think the sentence should be updated once they answer to (4).

Minor Comments 1. In the main text the authors indicate different appendix with capital letters, however, there are no letters in the appendix section.

  1. Introduction: “The motility of active particles is much higher than that of their passive counterparts” What does it mean? A passive bead immersed in a thermal bath is not motile. I think the authors mean that motility induces a diffusive regime whose diffusion constant is much bigger than that due to the thermal bath. I would ask the authors to state properly this sentence.

  2. Abstract: “Our results can be confirmed by real-space experiments on active colloidal Janus particles in the external field”. Usually, Janus particles are well captured by Active Brownian motion rather than AOUP,

  3. Introduction: “The ABP model is harder to use to make theoretical progress…” it does not look totally true to me, there are several theoretical works where the coarse-graining properties of active systems are obtained from ABP (see for instance T Speck, AM Menzel, J Bialké, H Löwen The Journal of chemical physics 142 (22), 224109) and RT (ME Cates, J Tailleur Annu. Rev. Condens. Matter Phys. 6 (1), 219-244) or even starting from minimal swimmer models (A Baskaran, MC Marchetti Proceedings of the National Academy of Sciences 106 (37), 15567-15572).

  4. Introduction: “AOUP model is recovered upon substituting $v_0^2=D_a/\tau$" I think in d dimensions (that is the situation where they are working in 2a and 2b, otherwise bold symbols do not make any sense) $D_a = v_0^2 \tau / d$.

  5. In section Model I think they start with considering AOUP in two spatial dimensions (since they write “\eta is a two-dimensional Ornstein-Uhlenbeck process”) however, I believe this information might get missed by a distracted reader. Could the authors introduce at the beginning of the section if they work in 1,2 or d spatial dimensions?

  6. Figure 1: why the linear force looks like $|x|$?

---

## Round 2 · Referee Report · Anonymous (Referee 2) · 2022-6-9

# Active particles driven by competing spatially dependent self-propulsion and external force

I thanks the authors for their answers to my concerns and for the changes implemented. I further apologize for the change of variable that I used to support my point about the similarity between spatially-dependent self-propulsion and driving forces: it contains indeed an error as the authors correctly noticed. I used it as a fast example to make my claim more precise but I should have detailed it more to exemplify what I meant. Let me correct it below by detailing how this change of variable should be carefully processed in order to map a dynamics with position-dependent self-propulsion speed onto one with a constant self-propulsion speed. I also take the opportunity to comment on the replies sent by the authors. All the references below correspond to the new version of the manuscript.

- The starting point is the evolution equation considered by the authors in one spatial dimension

$$\dot{x} = -\partial_x U + u(x)\eta \tag{1}$$

where $\eta$ is an OU noise. Assuming that the spatially-dependent activity $u(x)$ is a positive, bijective function then we can divide both sides by $u(x)$ to obtain

$$\frac{\dot{x}}{u(x)} = -\frac{\partial_x U}{u(x)} + \eta \ . \tag{2}$$

Then we define the primitive of $u^{-1}$ as $G(x) = \int_1^x 1/u(s)ds$. Because $u$ is a positive function, $G$ is also a bijection. Furthermore, we have that

$$\frac{dG(x)}{dt} = \frac{\dot{x}}{u(x)} \tag{3}$$

So, if we define the change of variable $\tilde{x} = G(x)$ we then have

$$\dot{\tilde{x}} = -\frac{\partial_x U(G^{-1}(\tilde{x}))}{u(G^{-1}(\tilde{x}))} + \eta \ , \tag{4}$$

which I think achieves to show that, in this case, one can map the model of the authors to the dynamics of an AOUP with a constant self-propulsion speed. This is only a rapid reasoning which might not be exempt of misstakes: I hope the authors will point out these caveats if they find them. However, if (4) is indeed confirmed, it really maps the

model of the authors to the dynamics of an AOUP with constant self-propulsion speed evolving in a complex (possibly non potential) landscape. And this is why I am not convinced by the authors' arguments supporting the motivation of their model. It seems to me that the effect of the interplay between spatially-dependent self-propulsion speed and potential forces will be equivalent, in a lot of cases and at least in one dimension (as studied in this paper), to the effect of a more complicated landscape in a normal setting with constant self-propulsion speed.

In addition, I don't understand the authors' description "non-Gaussianity induced by the interplay of confinement and spatially modulating swim velocity". As it is already known that an active particle with potential forces and constant self-propulsion speed has a non-Gaussian distribution both in $x$ and $v$, then there is no need to have an interplay with a spatially-dependent self-propulsion to get a non-Gaussian distribution.

- Concerning my remarks explaining how the findings of the authors on figure 1 and 2 could be retrieved by applying general and common principles of active matter, I understand the stance of the authors with respect to the accessibility of their manuscript to less experienced readers. However, I think that the Ref[74] (written by the authors), already provides an accessible manuscript with detailed numerics and analytics about a very similar problem. Furthermore, there are other references already tackling the problem of position-dependent self-propulsion in AOUPs such as Ref[63]. It is not clear to me what are the authors' additional contribution with respect to these previous works.

- In their replies, the authors pointed out a main difference between the model in Ref[63] and the model described in their manuscript, which, according to them, makes their proposed model more relevant for experiments. Indeed, they assert that the model in Ref[63] has a distribution of the form $\rho(x) = 1/u(x)$ (with $u(x)$ being once again the spatially dependent self-propulsion) only in a limited regime depending on the persistence length. However, I believe that this statement is wrong because a careful reading of Ref[63] shows that the distribution $\rho(\mathbf{r}) = 1/u(\mathbf{r})$ is recovered in a limited regime only when the self-propulsion speed $u(\mathbf{r})$ depends on the *set of position* $\{\mathbf{r}\}$ of the other active particles. When the self-propulsion speed $u(\mathbf{x})$ depends on the absolute position of only the particle under consideration, then the result $\rho(\mathbf{x}) = 1/u(\mathbf{x})$ of Ref[63] holds without any limitations. Thus, the authors have not convinced me about the relevance of their models with respect to the other ones in the literature. Why should it be relevant and more adequate for describing experiments?

- Finally, I would like to come back to the concern I expressed with respect to the authors' statement "the interplay between the external force and the modulation of the swim velocity can be used to manipulate the behavior of a confined active particle, for instance by locally increasing the kinetic temperature or by forcing the particles to accumulate in distinct spatial regions with different probability". My concern was that in their previous work [74] the authors have already shown that an active particle with spatially-dependent self-propulsion $u(x)$ has a distribution $\rho(x) \propto 1/u(x)$. Thus, my point was that the potential was not needed if one wanted to manipulate and sort active particles: a spatially-dependent $u(x)$ is enough.

  The authors reply that in a typical experimental setting the active particles are confined and that thus there is a need to study the interplay between confinement and self-propulsion speed.

  I am not convinced by this reply because if the particles are confined then one can just manipulate the confining potential in order to force the particles to accumulate where one wants. In my opinion, the authors did not clearly explain what are the physical features exhibited by their model which are not already present in an AOUP evolving in a complex potential with fixed self-propulsion speed (see the first point above with the change of variable). That is why I am not convinced by several claims of the authors about the emerging complex behaviour from the interplay between spatially-dependent self-propulsion and potential forces. For example, "the deviations of the velocity distributions from a Gaussian shape exclusively arise from the interplay of these two fields". For a single AOUP with fixed self-propulsion evolving in a (not harmonic) potential, the velocity distribution is already non gaussian without needing to invoke a complex interplay. Another example is the following statement "We demonstrated that by combining these two physically distinct effects, it is possible to generate complex density patterns through two relatively simple fields, which is surely easier to realize in practice than generating a single external field with a complex shape.". First, it is not clear to me why generating two external fields is easier than one, especially when this last one is a confining potential. I don't believe that it is hard to manufacture a sheet of plastic with some meanders and up-and-downs where you can put your active particles on. Second, it is not clear to me what is the aforementioned "complex density pattern" because, as I explained in my first review, one could have deduced it by applying two fundamental principles of active matter.

  Nonetheless, I agree with the authors that there are physical differences between an AOUP

with spatially-dependent self-propulsion speed and an AOUP in a confining potential which are clarified in the new footnote [76]. My concern is that I do not see the new complex behaviour or the new features due to the interplay between the two ingredients (potential forces and spatially-dependent self-propulsion) which are not already present whenever only one of the two ingredients is present.

My opinion is that the authors have not answered to my main concerns, and especially that the novelty and importance of their work with respect to the current literature (Ref[74] and Ref[63]) is not clear to me. Because of all the points discussed above, it seems to me that the main message of "interplay between spatially-dependent self-propulsion and potential forces yields complex physics" is not really grounded. Therefore, I would not recommend this manuscript for publication in scipost.

---

## Round 2 · Referee Report · Anonymous (Referee 3) · 2022-6-27

Report

Let me thank the authors for having considered my comments and for their replies. From my side, I have to say that they adequately addressed my major and minor comments. However, at his stage of the review process, I understand there are still open issues with Reviewer 2 (R2). It looks to me like there are two main criticisms that should be taken into account by the authors and addressed properly.

The first criticism raised by R2 is about the novelty of the work in comparison with (i) Ref [63] of the same authors, where they introduce the model considered here without any external fields, and (ii) Ref [74], where, in Section II-D (across p. 5 and 6), there is a short discussion about AOUPs in the presence of spatially varying activity in the small-tau limit. I think a clarification about that would be an additional improvement to the manuscript.

The second criticism looks to me to be still linked with point (i) of the previous comment: What is the new phenomenology one gets once we turn on a confining external potential in a system of AOUP with space-varying motility. As already shown in PRE 100, 052147 (2019) in the case of one-dimensional non-interacting run-and-tumble particles with space-dependent speed (see Eq. (7) of that reference), R2 noticed a possible mapping into the dynamics of AOUP in a complicated force field. My feeling is that, since the problem does not map simply into the dynamics with an effective conservative field, I think it might be opportune to talk about some sort of complex behavior for the resulting dynamics (if I interpret correctly the reply of the authors). Again, although the authors in their reply wrote about this issue showing that the dynamics is not just the dynamics of an active particle into an effective potential, I think it is opportune to have further improvements. For instance, the authors might better clarify how the competition between external confining potential and space-varying velocity works, what are the novelties with respect to the case without an external field, and why it is important to consider the effect of external fields.

---

## Round 2 · Referee Report · Anonymous (Referee 1) · 2022-7-4

Report

I thank the authors for their careful consideration of my comments and for modifying their manuscript accordingly. From my side, the authors have satisfactorily addressed all my concerns.

I have read the discussion between the authors and the second referee. The first main concern of the referee is that the dynamics of the AOUP with both a confining potential and a spatially-varying self-propulsion velocity can be mapped, under a change of variable, unto a dynamics with fixed self-propulsion velocity and an effective force field. The second main concern is the experimental relevance of using both a confining potential and a space-dependent self-propulsion velocity.

Given that the mapping proposed by the referee is rather convoluted, one could argue that the specific study of the interaction between an external potential and a varying self-propulsion velocity is relevant if there exist experimental realisations of these. I would suggest the authors to insist on this point.

To address the first concern, the authors may consider showing a specific example where the proposed mapping fails, e.g. because u(x) is not bijective, or partially explains the physics at play.

In the footnote [76], it is unclear why the physics is different: the particle escaping to infinity in the case of a modulation of the self-propulsion velocity is also a consequence of fluctuations induced in the active force. If the stationary density profile is identical, what quantity would distinguish the physics at play in the two systems considered?

---

## Round 2 · Author Response

The reply to the referees has been included in the pdf attached, before the new version of the paper.

---

## Round 2 · List of Changes

The list of changes is included in the pdf "2022June7_maintext_reply.pdf" together with the reply to the referees.

---

## Round 3 · Author Response

The reply to the referees is included in the pdf attached, which contains also the main text of the paper.

---

## Editorial Decision

published